# Assessing stratospheric contributions to subseasonal predictions of precipitation after the 2018 SSW from SNAPSI

Ying Dai[1], Peter Hitchcock[1], Amy H. Butler[2], Chaim I. Garfinkel[3], and William J. M. Seviour[4]

[1]Department of Earth and Atmospheric Sciences, Cornell University, Ithaca, NY, USA
[2]NOAA Chemical Sciences Laboratory, Boulder, CO, USA
[3]Fredy and Nadine Herrmann Institute of Earth Sciences, The Hebrew University of Jerusalem, Jerusalem, Israel
[4]Department of Mathematics and Statistics, University of Exeter, Exeter, United Kingdom

**Correspondence:** Ying Dai (yd385@cornell.edu)

**Abstract.** The sudden stratospheric warming (SSW) event in February 2018 was followed by dry spells in Scandinavia and record-breaking rainfall over the Iberian Peninsula through the following March. Here, we study the role of the 2018 SSW in subseasonal to seasonal (S2S) prediction of the 'Wet Iberia and Dry Scandinavia' precipitation signal, using a new database of S2S forecasts generated by the Stratospheric Nudging And Predictable Surface Impacts (SNAPSI) project. This database includes three sets of forecast ensembles: a free ensemble in which the atmosphere evolves freely, a nudged ensemble in which the stratosphere is nudged to the observed zonal-mean evolution of the 2018 SSW, and a control ensemble in which the stratosphere is nudged to climatology. Each set of ensembles has two initialization dates: 25 January 2018 and 8 February 2018, both before the onset of the SSW on 12 February. We find that the 'Wet Iberia and Dry Scandinavia' pattern was captured by the late free ensemble (initialized at 8 February) which successfully predicted the stratospheric warming, but not by the early free ensemble (initialized at 25 January) which predicted a stratospheric cooling. Unlike the early free ensemble, the early nudged ensemble successfully captured the 'Wet Iberia and Dry Scandinavia' pattern, indicating that an accurate forecast of stratospheric variability can improve S2S predictability of precipitation. While the pattern of European precipitation anomalies is evidently connected to the stratosphere, we estimate that only roughly a quarter of the amplitude was expected given the stratospheric anomalies. Nonetheless, Iberian rainfall extremes of equal strength or stronger than the one observed are twice as likely in the nudged ensemble than in the control ensemble. The increased likelihood in the nudged ensemble suggests that the weakened stratospheric polar vortex can increase the risk of Iberian rainfall extremes.

## 1  Introduction

Sudden stratospheric warmings (SSWs) are well-recognized as an important source of predictability for surface weather and climate on subseasonal to seasonal (S2S) timescales (Baldwin and Dunkerton, 2001; Thompson et al., 2002; Charlton and Polvani, 2007; Kolstad et al., 2010; Sigmond et al., 2013; Hitchcock et al., 2013; Hitchcock and Simpson, 2014; Butler et al., 2017; Domeisen et al., 2020; Rao et al., 2020). On average, periods following SSWs are associated with a persistent negative phase of the North Atlantic oscillation (NAO), which can lead to significant anomalies in surface air temperature and precipitation. Over the past two decades, it has been robustly demonstrated that SSW events are associated with increased likelihood

and severity of cold air outbreaks in Eurasia (Kolstad et al., 2010; Kretschmer et al., 2018; Huang and Tian, 2019; Huang
et al., 2021, 2022) and increased precipitation over Europe's Mediterranean coast (Butler et al., 2017; Ayarzaguena et al.,
2018; Domeisen and Butler, 2020). While these canonical impacts of SSWs are clear in the composite average, they are not
necessarily seen following individual SSW events, due to the large event-to-event variability in the tropospheric responses to
SSWs (Hall et al., 2021; Nebel et al., 2024).

The 2018 SSW event, which occurred on 12 February 2018, provides a clear example of the canonical tropospheric impact
of SSWs (Greening and Hodgson, 2019; Butler et al., 2020). In this specific event, the stratospheric warming was followed by
a persistent negative phase of the NAO, a long-lasting intense cold spell over Eurasia (Karpechko et al., 2018; Huang et al.,
2022), and extraordinarily rainy conditions over the Iberian Peninsula (Ayarzaguena et al., 2018). Due to its severe weather
impacts, the 2018 SSW event received special attention from the public; for instance, during several weeks following the 2018
SSW event, icy winds from Siberia blew across the European continent, which were called the 'Beast from the East' by the
media. The so-called 'Beast from the East' brought freezing temperatures, heavy snow, and strong winds to many regions
across Europe, along with power cuts, water supply problems, travel disruption, school closures, and weather-related deaths.
Moreover, the widespread disruption and danger to life caused by SSW-driven severe winter weather will likely occur again
because SSW events occur on average twice every three boreal winters. Therefore, a skillful forecast of an SSW event and the
extreme tropospheric state it brings is of great importance for issuing early weather warnings to the public when another SSW
event (like the 2018 event) and the impacts it brings are on their way. There is thus a clear need to evaluate the capabilities
of state-of-the-art operational S2S models to predict such an SSW and the extreme tropospheric state after it. The 2018 SSW
event provides an excellent case study with which to conduct the evaluation (Karpechko et al., 2018; Rao et al., 2020; Butler
et al., 2020).

For example, Karpechko et al. (2018) has evaluated the ability of S2S models to predict the 2018 SSW event and the pro-
longed cold snap after it. Analyzing an ensemble of nine subseasonal forecast models from the S2S Database (http://www.s2sprediction.net/),
Karpechko et al. (2018) found that four days before the onset date, all models predicted the SSW event with a very high cer-
tainty; the following long-lasting cold anomaly across Eurasia was also predicted, albeit with an underestimated magnitude.
However, they did not look into the predictability of the precipitation signal following the 2018 SSW. Another study by Rao
et al. (2020) analyzed the prediction of the rainfall anomalies following the 2018 SSW in S2S predictions from 11 models
participating in the S2S project. However, they did not check the precipitation signal over the Iberian Peninsula, where a
record-breaking precipitation event occurred after the 2018 SSW (Ayarzaguena et al., 2018). In addition, they did not validate
the precipitation signal using precipitation observations. In fact, very few studies of the impacts of SSWs on precipitation have
looked directly at precipitation observations; even the most 'robust' precipitation signal following SSWs–a 'Wet Iberia and Dry
Scandinavia' pattern–is identified in global models and reanalyses, but has not been confirmed in observations (Butler et al.,
2017; Ayarzaguena et al., 2018; Domeisen and Butler, 2020; Dai and Hitchcock, 2021; Dai et al., 2023, 2024).

In this study, we will investigate the S2S predictability of the precipitation signal after the 2018 SSW, with a focus on the
extreme precipitation over the Iberian Peninsula. In particular, we address the following questions: (1) is the precipitation signal
following the 2018 SSW identifiable in observations of precipitation? (2) is this precipitation signal predictable to some extent

on S2S timescales? (3) can a successful forecast of the 2018 SSW itself lead to more accurate forecasts of the precipitation signal?, and (4) to what extent did the 2018 SSW raise the probability of the extreme Iberian precipitation event?

To address the first question, we will use NASA satellite observations and GCOS Surface Network (GSN) in situ station observations of precipitation to identify the precipitation signal following the 2018 SSW. To address the rest of the questions, we will exploit a new database of S2S forecasts generated by the Stratospheric Nudging And Predictable Surface Impacts (SNAPSI) project. SNAPSI is a model intercomparison project aimed at assessing the role of SSWs in S2S forecasts, which consider three recent SSW events including the 2018 SSW (Hitchcock et al., 2022). For each target SSW event, the SNAPSI experimental design includes three forecast ensembles: a standard, unperturbed free ensemble, and two ensembles in which the zonal mean state of the stratosphere is nudged: a nudged-to-observations ensemble relaxed towards the observed evolution of the target SSW, and a nudged-to-climatology ensemble relaxed towards climatology. By construction, the impact of the 2018 SSW on S2S forecast can be identified by comparing the two nudged ensembles; the contribution of a successful forecast of the 2018 SSW to near-surface S2S forecast skill can be quantified by comparing the nudged-to-observations ensemble to the free ensemble; and the capabilities of S2S models to predict the 2018 SSW event and the subsequent tropospheric anomalies can be evaluated by comparing the free ensemble to high-quality satellite and in situ station observations and modern reanalysis products.

The data and methods used in this study are outlined in Section 2, the precipitation signal in observations is identified in Section 3, and the subseasonal predictability of the precipitation signal in SNAPSI models is investigated in Section 4, with the processes that lead to predictability in the precipitation signal discussed in Section 5. To help understand the stratospheric contribution to subseasonal predictability of the precipitation signal, a systematic component of precipitation associated with stratospheric variability is identified in Section 6. The role of the 2018 SSW in driving the extreme Iberian precipitation is evaluated in Section 7. The timing of the occurrence of the extreme Iberian precipitation is discussed in Section 8. The ensemble spread of individual SNAPSI models is discussed in Section 9. In Section 10 we provide a summary of this study.

## 2 Data and Methods

### 2.1 Satellite and in situ station observations of precipitation

For satellite observations of precipitation, we make use of the daily accumulated precipitation totals from NASA's Integrated Multi-satellitE Retrievals for Global Precipitation Measurement (IMERG, version 05), level-3 final run data product (https://gpm.nasa.gov/data/imerg). This dataset is available from June 2000 up to near real-time, and has a $0.1° \times 0.1°$ latitude-longitude resolution.

For in situ station observation of precipitation, we make use of the total daily precipitation from the Global Climate Observing System (GCOS) Surface Network (GSN) of in situ stations (https://gcos.wmo.int/en/networks/atmospheric/gsn). More than a hundred GSN stations offer data records that extend back to before the year 1900.

**Table 1.** SNAPSI datasets to be discussed.

| Participating center | Model |
|:---:|:---:|
| CCCma | CanESM5 |
| NCAR | CESM2-CAM6 |
| Meteo-France | CNRM-CM61 |
| CNR-ISAC | GLOBO |
| UKMO | GloSea6 |
| KMA | GloSea6-GC32 |
| SNU | GRIMs |
| ECMWF | IFS |

## 2.2 SNAPSI data sets

We use the daily output from 8 models (Table 1) participating in the SNAPSI project. At the time of writing, these are all the models available that provide the daily output of the fields examined in this study. These fields include precipitation (pr), sea level pressure (psl), 100-hPa air temperature ($T_{100}$), specific humidity (hus), eastward wind ($ua$), and northward wind ($va$). The last three quantities that depend on the vertical coordinate are available at 34 pressure levels from $1000\,\mathrm{hPa}$ up to $0.4\,\mathrm{hPa}$, including 1000, 925, 850, 700, 600, 500, 400, 300, 250, 200, 170, 150, 130, 115, 100, 90, 80, 70, 60, 50, 40, 30, 20, 15, 10, 7, 5, 3, 2, 1.5, 1.0, 0.7, 0.5, and $0.4\,\mathrm{hPa}$.

For each model, two initialization dates for the 2018 SSW event are considered: 25 January 2018 and 8 February 2018. For each initialization date, three sets of forecast ensembles are analyzed (Table 2). For the sake of simplicity, in the following, the forecast ensembles initialized at 25 January 2018 and 8 February 2018 will be referred to as the 'early' forecast ensemble and the 'late' forecast ensemble, respectively. Each set of forecast ensembles listed in Table 2 has an ensemble size of 50 members, and each member is a 45-day-long integration. The 45-day-long integration from the early and late forecast ensemble covers days $[1,26]$ and $[1,40]$ after the onset of the 2018 SSW, respectively. In this study, we will focus on days $[1,25]$ after the onset of the 2018 SSW, because days $[1,25]$ is the SSW-aftermath phase covered by the 45-day-long integration from both the early and late forecast ensembles. We use days $[1,25]$ rather than days $[1,26]$ because the former allows for the calculation of a 5-day average (see Fig. 10 below). This 25-day period corresponds roughly to weeks 4 through 6 of the early forecast ensemble and weeks 2 through 4 of the late forecast ensemble.

For the comparison across different models and the calculation of the multi-model-ensemble (MME) mean, data are linearly interpolated to a common grid at a $1.0° \times 1.0°$ latitude-longitude resolution.

**Table 2.** Forecast ensembles to be discussed for each model listed in Table 1. For both nudged and control ensembles, the nudging region has a lower limit of 90 hPa.

| Forecast ensemble | Experimental design |
|---|---|
| free | A standard forecast ensemble in which the atmosphere evolves freely after initialization. |
| nudged | A nudged ensemble in which the zonal-mean stratospheric state is nudged globally to the observed time evolution of the 2018 SSW event. |
| control | A nudged control ensemble in which the zonal-mean stratospheric state is nudged globally to a time-evolving climatological state. |

## 2.3 Reanalysis data sets

We use daily data from the ERA5 reanalysis dataset (Hersbach et al., 2020) archived at a $1.0° \times 1.0°$ latitude-longitude resolution and 37 pressure levels from 1000 hPa up to 1 hPa. The 37 pressure levels are: 1000, 975, 950, 925, 900, 875, 850, 825, 800, 775, 750, 700, 650, 600, 550, 500, 450, 400, 350, 300, 250, 225, 200, 175, 150, 125, 100, 70, 50, 30, 20, 10, 7, 5, 3, 2, and 1 hPa.

## 2.4 Definition of anomalies

For IMERG, GSN, and ERA5 precipitation, daily anomalies are calculated by subtracting the daily climatology on that calendar day. The daily climatology is computed as the multi-year average for each calendar day.

For the rest ERA5 variables, daily anomalies are obtained by subtracting the seasonal cycle on that calendar day. The seasonal cycle is defined as the mean and first three Fourier harmonics of the daily climatology to remove the high-frequency sampling variability, following previous studies (Compo and Sardeshmukh, 2004; Simpson et al., 2013; Ogrosky and Stechmann, 2015).

Note that for the precipitation field, the anomalies are calculated with reference to the daily climatology, rather than the seasonal cycle. This is because observational precipitation like GSN have missing values on certain days and at certain locations. These missing values may result in gaps in the daily climatology at certain locations. The presence of gaps in the daily climatology precludes the use of smoothing techniques like the Fourier filter.

For the majority of the analysis (except Section 6) in this study, the daily climatology is computed over the period 2000-
2020. Such a baseline period is determined by the availability of the IMERG data, since the GSN and ERA5 data are available over much longer time periods.

For the analysis in Section 6, the daily climatology is computed over the period 1950–2022. This is because Section 6 involves an inter-SSW regression analysis that uses all 46 historical SSW events during the 1950-2022 period from the ERA5 reanalysis (see Section 6 for more details about the regression analysis).

With regard to the SNAPSI ensembles, the anomalies in the free and nudged ensembles are obtained by calculating the differences from the corresponding control ensemble.

## 2.5    Water Vapor Transport

For both SNAPSI and ERA5 datasets, we calculate the vertically integrated water vapor transport (IVT), which is directly and closely related to precipitation (Rutz et al., 2014; Guan and Waliser, 2015). Following these studies, IVT is defined as:

$$\mathrm{IVT}_x = \frac{1}{g} \int_{1000\,\mathrm{hPa}}^{100\,\mathrm{hPa}} qu\,\mathrm{d}p, \tag{1}$$

$$\mathrm{IVT}_y = \frac{1}{g} \int_{1000\,\mathrm{hPa}}^{100\,\mathrm{hPa}} qv\,\mathrm{d}p, \tag{2}$$

$$\mathrm{IVT} = \sqrt{\mathrm{IVT}_x^2 + \mathrm{IVT}_y^2}, \tag{3}$$

where $\mathrm{IVT}_x$ and $\mathrm{IVT}_y$ denote the vertical integral of eastward and northward water vapor flux (units: $\mathrm{kg\,m^{-1}\,s^{-1}}$), respectively, $g$ is the acceleration of gravity ($\mathrm{m\,s^{-2}}$), $p$ means pressure (units: $\mathrm{hPa}$), $q$ represents specific humidity (units: $\mathrm{kg\,kg^{-1}}$), and $u$ ($v$) is the eastward (northward) component of wind (units: $\mathrm{m\,s^{-1}}$). The integration is done using 12 pressure levels between $1000\,\mathrm{hPa}$ and $100\,\mathrm{hPa}$, including 1000, 925, 850, 700, 600, 500, 400, 300, 250, 200, 150, and $100\,\mathrm{hPa}$. These 12 pressure levels are shared by SNAPSI and ERA5 datasets, enabling a direct comparison of IVT between SNAPSI and ERA5.

## 145  3    Observational evidence of precipitation signals after the 2018 SSW

In this section, we use NASA satellite observations and GSN in situ station observations of precipitation to identify precipitation signals after the 2018 SSW, and then compare them to those in ERA5 precipitation. The inclusion of observational precipitation adds value by providing an independent and often more accurate reference, helping to validate and complement ERA5 precipitation that does not directly assimilate any rain-gauge data (Lavers et al., 2022). It turns out that the precipitation

signals after the 2018 SSW agree well between observations and the reanalysis, regarding both the time-mean features (Fig. 1) and the temporal evolutions (Fig. 2).

Regarding the time-mean features (Fig. 1), the 2018 SSW is followed by dry spells over Scandinavia (blue box in each panel) and rainy conditions over Iberian (red box in each panel). Between Scandinavia and Iberia, the British Isles underwent a 'Southern Wet and Northern Dry' condition (black box between the blue and red boxes). These features over Europe are

robustly identifiable in IMERG (Fig. 1a), GSN (Fig. 1b), and ERA5 (Fig. 1c). The opposite-signed changes between northern and southern Europe after the 2018 SSW are consistent with the composite-mean precipitation response to historical SSWs

both in the reanalysis [e.g., Fig. 4c in Butler et al. (2017)] and in global climate models [e.g., Fig. 11 in Dai et al. (2024)]. In addition, the 2018 SSW was followed by anomalously dry conditions over southern Greenland, which are also consistent with the composite-mean precipitation response to historical SSWs. Besides these canonical features, the 2018 SSW is followed by unique precipitation signals that are not seen in the composite-mean precipitation response to historical SSWs. For example, the 2018 SSW is followed by anomalously dry conditions over eastern Canada and anomalously wet conditions over the central United States (Fig. 1). Again, all these features in North America are identifiable in IMERG (Fig. 1a), GSN (Fig. 1b), and ERA5 (Fig. 1c). However, it is unclear whether these North American precipitation signals are attributable to the 2018 SSW, since historical SSWs, on average, are followed by weak and insignificant precipitation signals over eastern Canada and the central United States [e.g., Fig. 4c in Butler et al. (2017); Fig. 11 in Dai et al. (2024)]. These time-mean features of the precipitation signals averaged over days [1,25] discussed above are not sensitive to the averaging period. Qualitatively similar results are obtained using days [1,40] to calculate the time-mean precipitation signals (Fig. S1).

The regions considered here exhibit distinct temporal evolution. For example, the dry spells over Scandinavia appeared almost right after the onset of the 2018 SSW (Fig. 2a, thick curves) whereas the rainy conditions over Iberia did not appear until 15 days after the onset of the 2018 SSW (Fig. 2b, thick curves). After their appearance, the dry spells over Scandinavia underwent insignificant synoptic-scale fluctuations (Fig. 2a, thin curves) while the rainy conditions over Iberia experienced three large peaks (Fig. 2b, thin curves). The first peak occurred in late February and early March 2018, when Winter Storm Emma brought heavy snow to Europe. These distinct features can be seen in all of the three datasets considered here, including IMERG (purple curves), GSN (green curves), and ERA5 (gray curve). Note that amongst the three datasets, IMERG tends to have a stronger precipitation signal than the other two (Fig. 2 and Fig. S2a,c, thick curves).

The above results provide observational evidence of the precipitation signals after the 2018 SSW, as indicated by the agreement between observations and the reanalysis over the widespread area shown in Fig. 1. In the following, our analysis will focus on the dry spells over Scandinavia and the rainy conditions over Iberia, which are canonical precipitation responses to SSWs, in association with an equatorward shift of the North Atlantic storm track (Butler et al., 2017; Domeisen and Butler, 2020; Dai and Hitchcock, 2021; Afargan-Gerstman et al., 2024).

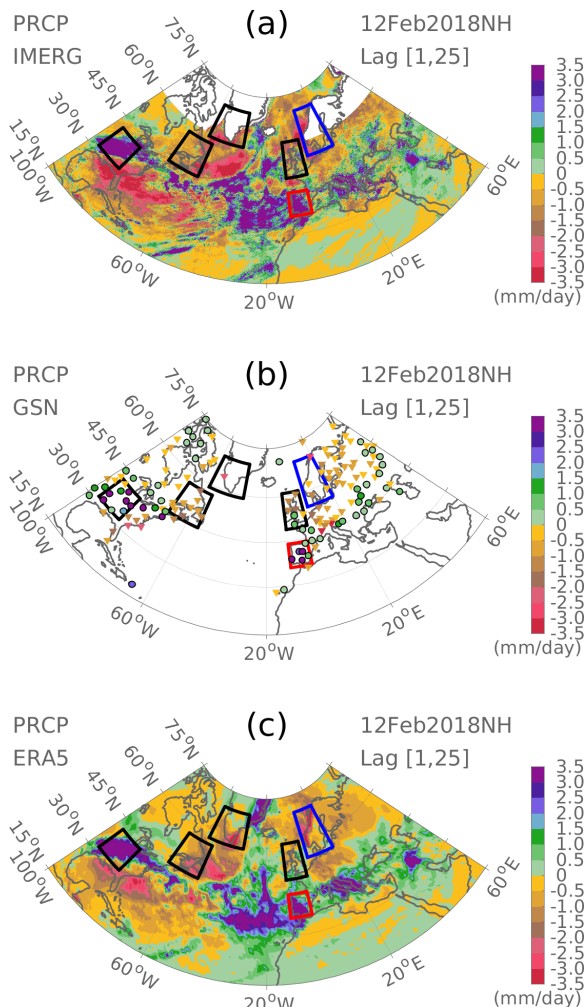

**Figure 1.** Observed precipitation anomalies averaged over lag days [1,25] relative to the SSW onset date from (a) IMERG and (b) GSN. (c) Precipitation anomalies averaged over the same period from ERA5. The three black boxes to the left of $20°$W, from top to bottom, denote Greenland ($59°$N-$70°$N, $58°$W-$35°$W), Gulf St Lawrence ($45°$N-$57°$N, $70°$W-$54°$W), and Mississippi Plain ($32°$N-$42°$N, $96°$W-$84°$W), respectively. The three boxes to the right of $20°$W, from top to bottom, denote Scandinavia ($55°$N-$70°$N, $4°$E-$20°$E), British lsles ($49°$N-$61°$N, $10°$W-$1°$E), and Iberia ($36°$N-$44°$N, $10°$W-$1°$W), respectively. These boxes indicate domains to be discussed in Fig. 2. Note that IMERG precipitation in panel a provides only partial spatial coverage at latitudes above $60°$. This is because infrared-based precipitation estimates cannot be included at higher latitudes, so the coverage is limited to grid boxes for which there is no snow/ice on the surface.

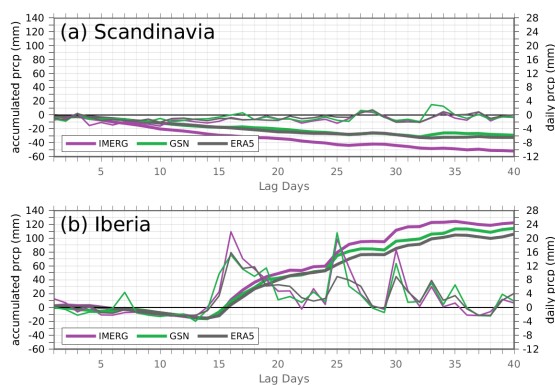

**Figure 2.** Observed precipitation anomalies averaged over (a) Scandinavia and (b) Iberia from IMERG (purple curves) and GSN (green curves) over lag days $[1, 40]$ relative to the SSW onset date. Precipitation anomalies over the same period from ERA5 are shown in gray curves. Thick curves and the left $y$-axis correspond to the accumulated precipitation anomalies; thin curves and the right $y$-axis correspond to the daily precipitation anomalies. These two domains Scandinavia and Iberia are indicated by the blue and red boxes in Fig. 1, respectively.

## 4  Subseasonal predictability of precipitation signals after the 2018 SSW

After demonstrating the robustness of precipitation signals after the 2018 SSW, we now investigate the subseasonal predictability of these precipitation signals using the S2S forecasts generated by the SNAPSI project.

We start by demonstrating the predictability of the time-mean features of precipitation signals. For this purpose, we first evaluate the multi-model ensemble mean result (Fig. 3). As can be seen, the 'Wet Iberia and Dry Scandinavia' pattern in the reanalysis (Fig. 3a) is well captured in the early nudged ensemble (Fig. 3b). Initialized at the same date, the early free ensemble predicts an opposite-signed 'Dry Iberia and Wet Scandinavia' pattern (Fig. 3c). The improved precipitation forecast by the early nudged ensemble relative to the early free ensemble (Fig. 3b vs. Fig. 3c) suggests that a successful stratospheric forecast has the potential to improve near-surface forecast skill for precipitation signals following the SSW event. Initialized later at 8 February 2018, both the nudged ensemble and the free ensemble capture the observed 'Wet Iberia and Dry Scandinavia' pattern (Fig. 3d,e). We will show later that this is because the late free ensemble successfully captures the sudden warming in the stratosphere. Note that the ensemble mean magnitude of precipitation signals is much smaller than that in ERA5, regardless of the initialization dates or the application of the nudging technique [note the different color bars between panel (a) and the rest panels of Fig. 3]. The magnitude of the ensemble mean signal will be examined in Section 6.

Among the eight models analyzed in this study, most models predict precipitation signals that are qualitatively similar to each other (Figs. S3-S6). For example, the nudged ensemble of each model captures the observed 'Wet Iberia and Dry Scandinavia' pattern, regardless of the initialization date (Figs. S3 and S5). This is not surprising because the nudged ensemble of each model provides a successful forecast of stratospheric variability by design. Regarding the free ensembles, initialized at 25 January 2018, most models predict a 'Dry Iberia and Wet Scandinavia' pattern (Fig. S4), which is similar to the multi-model ensemble mean (Fig. 3c) but opposite from the reanalysis (Fig. 3a); One exception is CNRM-CM61, which captures the observed 'Wet

Iberia and Dry Scandinavia' pattern (Fig. S4e). Initialized on 8 February 2018, the free ensemble of most models captures the observed 'Wet Iberia and Dry Scandinavia' pattern (Fig. S6); One exception is GloSea6-GC32, which predicts dry conditions over Iberia (Fig. S6d). Despite the general agreement on the sign of precipitation signals across models, the magnitude of precipitation signals varies from model to model (Figs. S3-S6). Furthermore, none of the eight models capture the observed magnitude of precipitation signals: the forecasted precipitation signals from each model are much weaker than the reanalysis [note the different color bars between Fig. 3a and Figs. S3-S6]. As a result, the multi-model ensemble mean precipitation signals (Fig. 3b-e) are much weaker than the reanalysis (Fig. 3a).

We now evaluate the predictability of the temporal evolution of the precipitation signals. As shown in Fig. 2a, the observed dry condition over Scandinavia occurred right after the onset of the 2018 SSW with insignificant synoptic-scale fluctuations thereafter. These observed features are well captured by the two nudged ensembles and the late free ensemble (Fig. 4a, green, gray, and purple curves), albeit with a much weaker magnitude than the reanalysis (note the different scales of y-axis between Fig. 2 and Fig. 4). Regarding the observed wet conditions over Iberia, its temporal evolution is poorly forecasted by all forecast ensembles. In the observations, the wet conditions over Iberia did not appear until 15 days after the onset of the 2018 SSW (Fig. 2b). By contrast, the forecasted wet conditions over Iberia occurred almost right after the onset of the SSW (Fig. 4b). This disagreement between forecasts and observations in terms of the time of occurrence of the wet conditions over Iberia will be discussed in Section 8. Note that the two nudged ensembles do not always agree with each other, especially for precipitation signals over North America (Fig. S7b and c, green vs. gray curves). Mindful that by construction, the two nudged ensembles have identical stratospheric states, the fact that their forecasted precipitation signals over North America differ from each other implies that the 2018 SSW may not play a dominant role in driving the precipitation signals over North America.

In summary, precipitation signals after the 2018 SSW feature a 'Wet Iberia and Dry Scandinavia' pattern. The spatial feature of such a pattern is successfully captured by the nudged ensembles initialized at both dates and the late free ensemble, but not captured by the early free ensemble. However, the forecasted magnitude of precipitation signals is only about one-fourth of that in observations. In addition, the wet Iberia conditions forecasted by all ensembles occur too early compared to observations. The magnitude and timing of the ensemble mean signal will be discussed in Section 6 and 8, respectively.

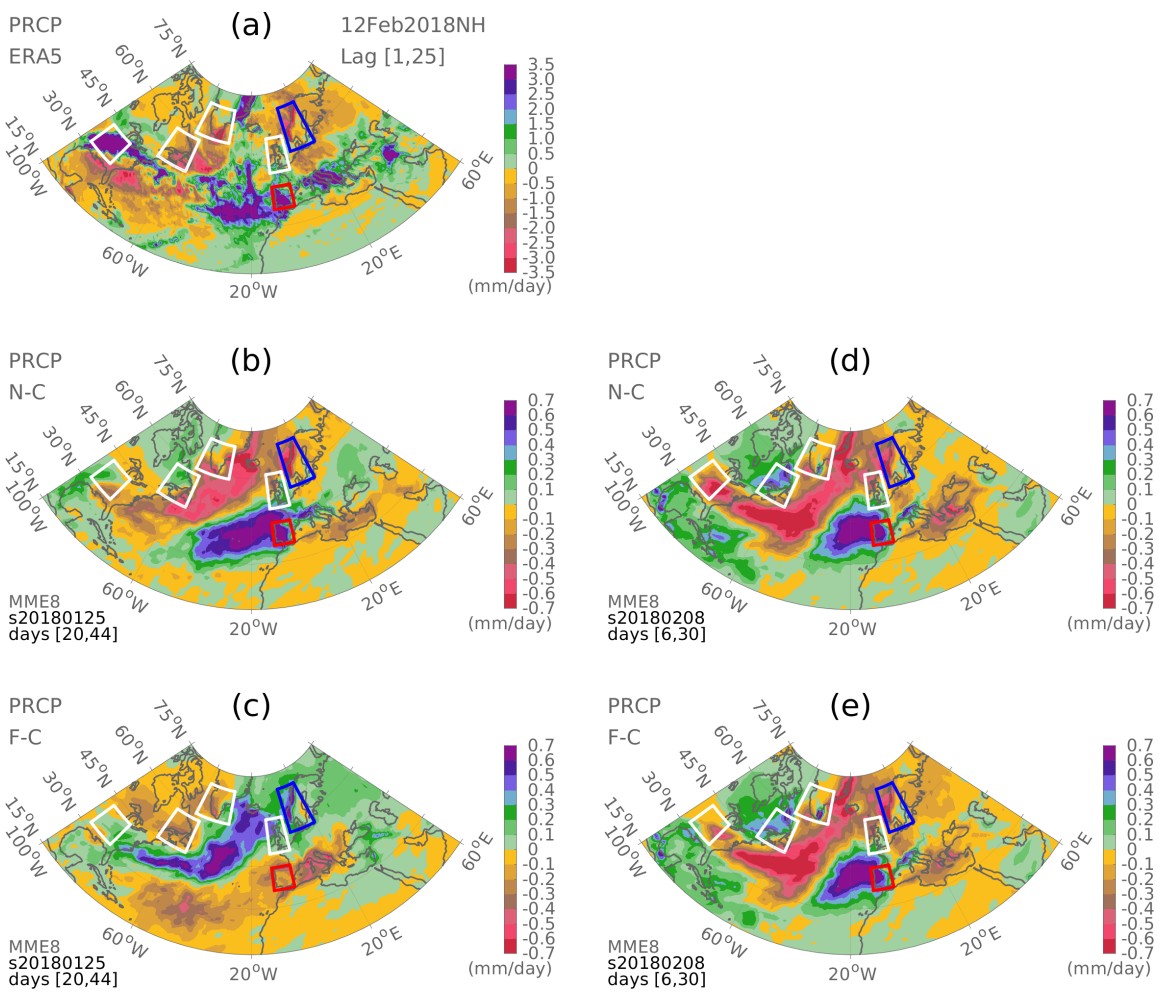

**Figure 3.** (a) Precipitation anomalies from ERA5, averaged over lag days [1,25] relative to the SSW onset date. Multi-model ensemble mean precipitation anomalies averaged over the same period from the (b) early nudged ensemble, (c) early free ensemble, (d) late nudged ensemble, and (e) late free ensemble. See Supplementary Figures 3-6 for the result of every single model. The six boxes here indicate the same domains as those in Fig. 1, except that the four black boxes in Fig. 1 are shown in white here.

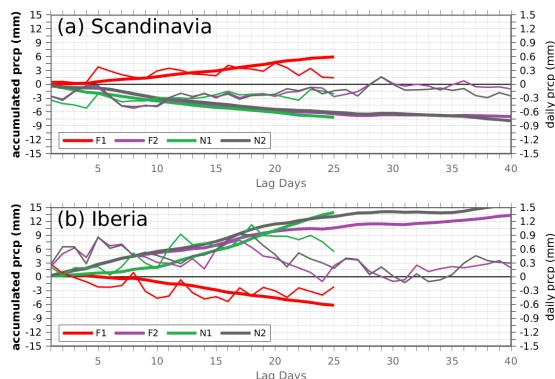

**Figure 4.** Multi-model ensemble mean precipitation anomalies averaged over (a) Scandinavia and (b) Iberia from the early free ensemble (red curves), late free ensemble (purple curves), early nudged ensemble (green curves), and late nudged ensemble (gray curves). Thick curves and the left $y$-axis correspond to the accumulated precipitation anomalies; thin curves and the right $y$-axis correspond to the daily precipitation anomalies. These two domains Scandinavia and Iberia are indicated by the blue and red boxes in Fig. 3, respectively.

## 5   Processes that lead to the predictability of precipitation signals after the 2018 SSW

After demonstrating the potential skill of S2S models in subseasonal forecasts of precipitation signals after the 2018 SSW, we now identify processes underlying this skill.

We first examine to what extent the sudden warming in the stratosphere is captured in the ensemble forecasts. To this end, for both SNAPSI and ERA5 datasets, the lower-stratospheric (100 hPa) polar cap (60°N-90°N) temperature anomaly ($\Delta T_{100}$) is calculated. Here the quantity $\Delta T_{100}$ is considered because it indicates the strength of the downward propagation of stratospheric anomalies (Charlton et al., 2007; Hitchcock et al., 2013; Karpechko et al., 2017) and has been identified as a major factor modulating the surface impact of SSWs (Dai et al., 2024). According to the reanalysis, the 2018 SSW is associated with a positive temperature anomaly of 4.8 K at 100 hPa (Table 3). This stratospheric warming is well captured in both nudged ensembles (Table 3), which is not surprising because their stratospheric state is nudged to follow the observed SSW. One may note that the lower-stratospheric warming at 100 hPa in the nudged ensembles is somewhat larger than, rather than identical to, that in the reanalysis. This might arise from the fact that 100 hPa is below the nudging region, whose lower limit is at 90 hPa (Hitchcock et al., 2022). This fact also explains why the inter-model spread of $\Delta T_{100}$ in the nudged ensembles is narrow but non-zero. The late free ensemble also captures the observed stratospheric warming but with a substantially larger inter-model spread than the nudged ensembles. Initialized earlier at 25 January 2018, the early free ensemble predicts a cooling in the stratosphere, which is opposite from the observed warming.

The lower-stratospheric temperature anomalies are associated with changes in the large-scale atmospheric circulation at the surface. In the reanalysis, the 2018 SSW is followed by a persistent negative NAO-like SLP anomaly (Fig. 5a), which is a canonical surface response to SSW. The negative NAO-like SLP pattern is captured in those ensembles that successfully capture the stratospheric warming, including the two nudged ensembles (Fig. 5b,d) and the late free ensemble (Fig. 5e). For

these three sets of ensembles, every single model captures the observed negative NAO-like SLP pattern, although the magnitude of the forecasted SLP anomaly varies from model to model (Figs. S8, S10, and S11). By contrast, a positive NAO-like SLP anomaly is predicted by the early free ensemble (Fig. 5c), which predicts a cooling in the stratosphere (Table 3). For the early free ensemble, seven out of eight models predict a positive NAO-like SLP anomaly (Fig. S9); One exception is CNRM-CM61, which captures the observed negative NAO-like SLP pattern (Fig. S9e). Note that CNRM-CM61 is also the only model

whose early free ensemble captures the observed 'Wet Iberia and Dry Scandinavia' pattern (Fig. S4e). Again, the magnitude of forecasted anomalies from all of the four ensembles is smaller than the observed magnitude (note the different color bars between panel (a) and the other panels of Fig. 5, as well as between Fig. 5a and Figs. S8-S11). The magnitude of the forecasted SLP anomaly (Fig. 5), together with the magnitude of the forecasted precipitation signals mentioned above (Fig. 3), will be discussed in the next section.

The large-scale atmospheric circulation can further affect the characteristics of IVT, which have a direct and close linkage to precipitation. In the reanalysis, in the presence of the negative NAO-like SLP anomaly, there is abundant moisture transport from the tropics to Iberia (Fig. 6a), resulting in extreme rainfall over Iberia (Fig. 3a, red box). Meanwhile, a lack of moisture transport to the north of Iberia (Fig. 6a) leads to dry conditions in Scandinavia (Fig. 3a, blue box). As a result, a 'Wet Iberia and Dry Scandinavia' pattern is seen after the 2018 SSW (Fig. 3a). In the two nudged ensembles and the late free ensemble that

captured the negative NAO-like SLP pattern (Fig. 5b,d,e), the forecasted IVT exhibits features similar to the reanalysis, with abundant moisture transported to Iberia and little moisture transported towards Scandinavia (Fig. 6b,d,e), which explains the 'Wet Iberia and Dry Scandinavia' pattern forecasted in the two nudged ensembles and the late free ensemble (Fig. 3b,d,e). For these three sets of ensembles, the forecasted IVT by most models exhibit qualitatively similar features to observations (Figs. S12, S14, and S15). With regard to the early free ensemble that predicts a positive NAO-like SLP pattern (Fig. 5c), there is

abundant moisture transport from the tropics to Scandinavia (Fig. 6c), resulting in excess rainfall over Scandinavia (Fig. 3c, blue box). Meanwhile, little moisture is transported over Iberia (Fig. 6c), leading to dry conditions in Iberia (Fig. 3c, red box). As a result, a 'Dry Iberia and Wet Scandinavia' pattern is forecasted by the early free ensemble (Fig. 3c). Again, CNRM-CM61 is the only exception among the eight models, whose early free ensemble captures the observed characteristics of IVT (Fig. S13e). This explains the 'Wet Iberia and Dry Scandinavia' pattern forecasted by CNRM-CM61's early free ensemble (Fig.

S4e).

In summary, the subseasonal predictability of precipitation signals after the 2018 SSW arises from a successful forecast of the sudden warming itself, which can further lead to a skillful forecast of the negative NAO-like SLP pattern and the large-scale characteristics of IVT. These results highlight the importance of getting a successful stratospheric forecast: with a successful forecast of the sudden warming, S2S models can capture the precipitation signals after the 2018 SSW several weeks in advance

(i.e., the early nudged ensemble). It is important to note that not all SSW events lead to a persistent negative NAO or European precipitation anomalies. This event-to-event variability in the surface response to SSWs introduces uncertainty into any predictive framework based solely on the occurrence of SSWs. One way to mitigate this uncertainty is by conditioning predictions on specific characteristics of the SSW (e.g., type, magnitude, and tropospheric precursors) (Maycock and Hitchcock, 2015; Kodera et al., 2016; Runde et al., 2016; de la Cámara et al., 2017; White et al., 2019; Xu et al., 2022).

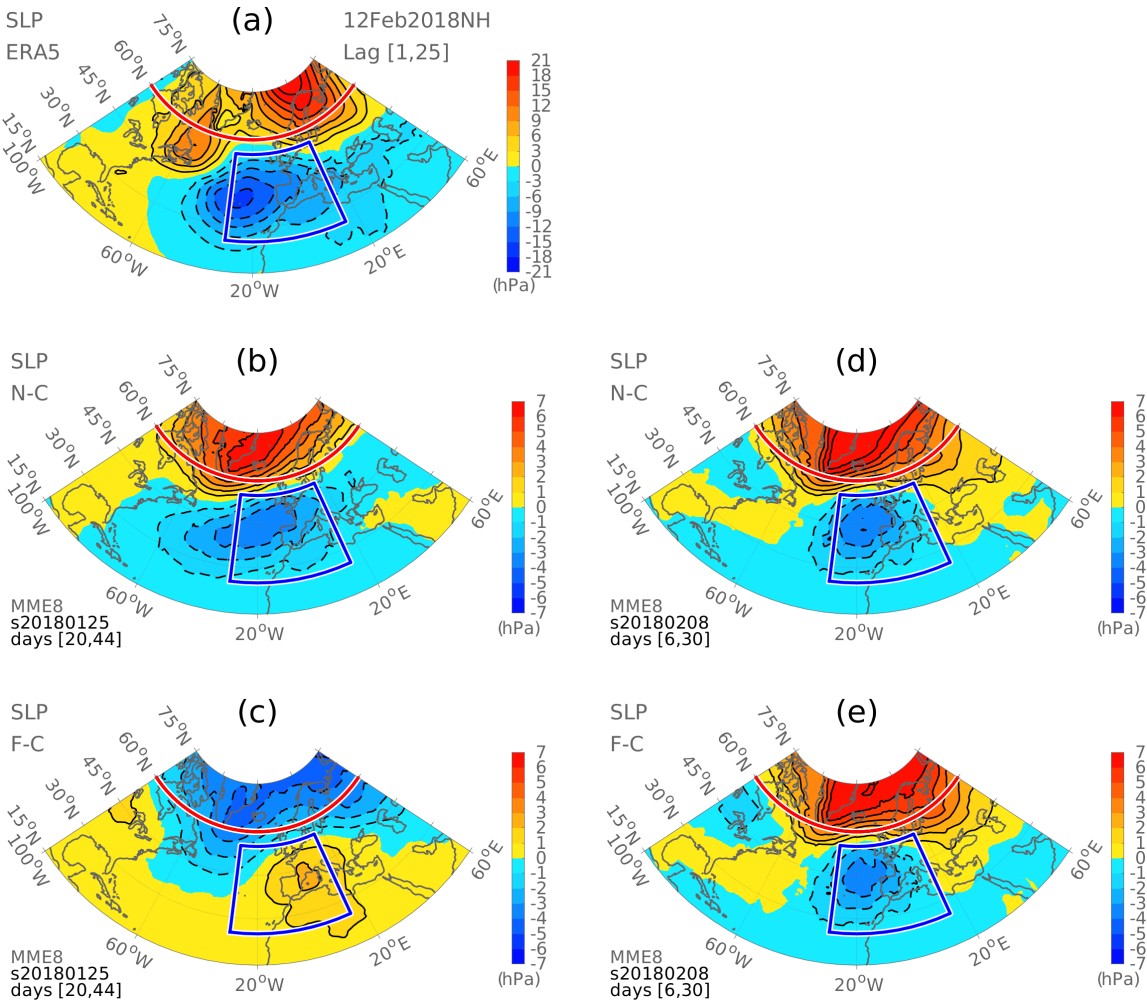

**Figure 5.** (a) SLP anomalies from ERA5, averaged over lag days [1,25] relative to the SSW onset date. Multi-model ensemble mean SLP anomalies averaged over the same period from the (b) early nudged ensemble, (c) early free ensemble, (d) late nudged ensemble, and (e) late free ensemble. See Supplementary Figures 8-11 for the result of every single model. The blue box indicates the domain ($25°$N-$55°$N, $30°$W-$15°$E) within which the Atlantic SLP response is calculated, which will be discussed in Table 3. The red curve indicates the latitude $60°$N to the north of which the Arctic SLP resposne is calculated, which will be discussed in Table 4. Note that the domain used to calculate the Arctic SLP response extends from $60°$N towards the North Pole ($90°$N), and spans the entire range of longitudes from $0°$ to $360°$, rather than being confined to the fan-shaped region over the Lambert projection used here.

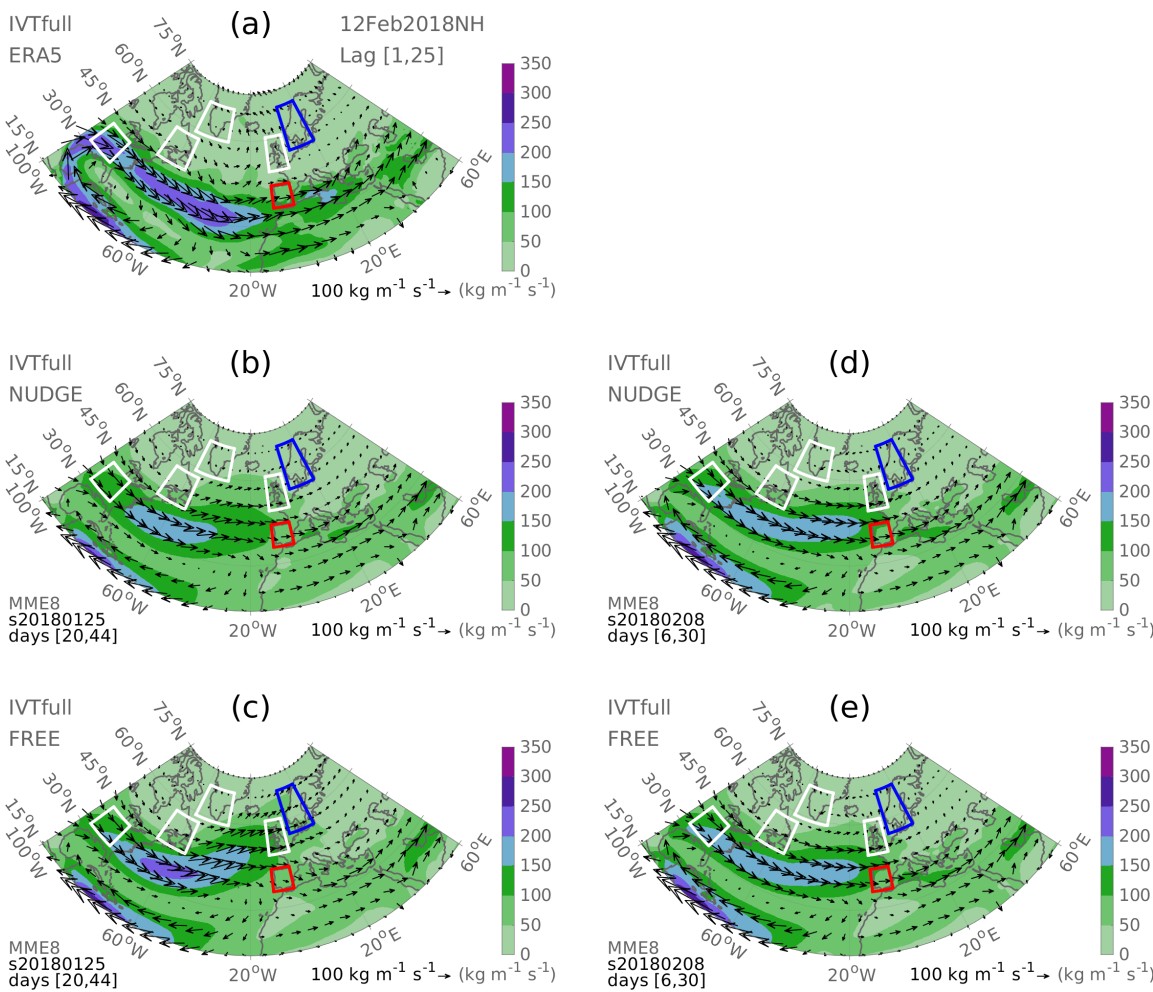

**Figure 6.** (a) Vertically integrated water vapor transport (IVT) field (shadings) and water vapor flux (vector) from ERA5, averaged over lag days [1,25] relative to the SSW onset date. Multi-model ensemble mean IVT and water vapor flux averaged over the same period from (b) early nudged ensemble, (c) early free ensemble, (d) late nudged ensemble, and (e) late free ensemble. See Supplementary Figures 12-15 for the result of every single model.

**Table 3.** Stratospheric anomalies and the mid-latitude tropospheric anomalies following the 2018 SSW. Here, $\Delta T_{100}$ is the area-weighted mean 100-hPa polar cap (60°N-90°N) temperature anomaly, $\Delta SLP_{Atlantic}$ is the area-weighted mean SLP anomaly within the blue box in Fig. 5, and $\Delta PRCP_{Iberia}$ is the area-weighted mean precipitation anomaly within the red box in Fig. 3. For ERA5, the full anomaly and its predictable component ($\pm$ 90% confidence intervals) are shown. For the nudged and free ensembles (whose anomalies are defined as differences from the corresponding control ensemble), the multi-model ensemble mean anomaly $\pm$ one standard deviation range across individual models is shown.

| 12-Feb-2018 SSW | Lag [1,25] | $\Delta T_{100}$ (K) | $\Delta SLP_{Atlantic}$ (hPa) | $\Delta PRCP_{Iberia}$ (mm/day) |
|---|---|---|---|---|
| ERA5 | full | 4.8 | -7.5 | 2.4 |
| | predictable | | -2.0±0.8 | 0.6±0.4 |
| s20180125 | nudged | 5.5±0.3 | -1.9±0.5 | 0.6±0.2 |
| | free | -5.7±3.7 | 0.9±1.7 | -0.3±0.4 |
| s20180208 | nudged | 5.8±0.4 | -1.7±0.9 | 0.5±0.2 |
| | free | 4.6±2.6 | -1.5±0.9 | 0.4±0.3 |

## 6 A systematic component of precipitation associated with stratospheric variability

The above results exhibit a notable feature: even for those ensembles that provide a successful forecast of the sudden warming in the stratosphere, the forecasted magnitude of tropospheric anomalies in the ensemble mean is substantially smaller than that observed. One possible explanation for this discrepancy (and one we will soon reject) is that the surface response to SSWs is too weak in these models. Alternately, this feature might arise because stratospheric variability is not the only factor determining the magnitude of observed tropospheric anomalies after the SSW. Other factors like tropospheric internal variability might also play a role. In order to sort through these various possibilities, this section will quantify the systematic component of tropospheric anomalies associated with stratospheric variability, and thereby demonstrate that i) the magnitude of the surface signal simulated by the models is, in fact, correct; and ii) internal variability likely played a large role in the unexpectedly large observed response.

For this purpose, a linear regression model is built between tropospheric anomalies and lower-stratospheric temperature anomalies using the 46 historical SSW events [defined by the reversal of the zonal mean westerlies at 60°N and 10-hPa following Charlton and Polvani (2007)] in the ERA5 reanalysis during the 1950-2021 extended winters. For each historical SSW event, the stratospheric-variability-associated component of tropospheric anomalies for some variable of interest ($\widehat{\Delta VAR}^i$) can be estimated by the regression equation:

$$\widehat{\Delta VAR}^i = \alpha_1 \times \Delta T_{100}^i + \alpha_0 \qquad (4)$$

Here, a linear regression model and the lower-stratospheric temperature anomalies are used because the tropospheric response to SSWs is found to be linearly governed by the strength of the lower-stratospheric warming (White et al., 2020, 2022). In the following, the stratospheric-variability-associated component of tropospheric anomalies estimated in equation 4 will be referred to as the 'predictable component' in the presence of a perfect knowledge of stratospheric variability.

The full tropospheric anomalies ($\Delta\mathrm{VAR}^i$) is the sum of the predictable component ($\widehat{\Delta\mathrm{VAR}}^i$) defined in equation 4 and the residual $\varepsilon^i$:

$$\Delta\mathrm{VAR}^i = \alpha_1 \times \Delta T_{100}^i + \alpha_0 + \varepsilon^i \tag{5}$$

In the above equations, the superscript '$i$' varies from 1 to 46, each indicating an individual SSW event from ERA5. We then use the above equations to estimate the predictable component of tropospheric anomalies after the 2018 SSW event. This
includes the Atlantic SLP anomaly ($\Delta\mathrm{SLP}_{\mathrm{Atlantic}}$) and the Iberian precipitation anomaly ($\Delta\mathrm{PRCP}_{\mathrm{Iberia}}$) in the mid-latitude (Table 3), as well as the Arctic SLP anomaly ($\Delta\mathrm{SLP}_{\mathrm{Arctic}}$) and the Scandinavian precipitation anomaly ($\Delta\mathrm{PRCP}_{\mathrm{Scandinavia}}$) in the subpolar-polar regions (Table 4).

As can be seen in Table 3, the predictable component of $\Delta\mathrm{SLP}_{\mathrm{Atlantic}}$ is estimated to be -2.0 hPa, which is about one-fourth of the full $\Delta\mathrm{SLP}_{\mathrm{Atlantic}}$ in ERA5 (-7.5 hPa). This is also the case for $\Delta\mathrm{PRCP}_{\mathrm{Iberia}}$, whose predictable component (0.6 mm/day)
is about one-fourth of the full anomaly (2.4 mm/day). These results suggest that the mid-latitude surface anomalies after the 2018 SSW arise largely from tropospheric internal variability, with only about one-fourth of the tropospheric anomalies arising from stratospheric variability. This explains why the magnitude of mid-latitude surface anomalies in SNAPSI ensemble means is substantially smaller than that in ERA5: even for those ensembles that successfully captured the sudden warming in the stratosphere (e.g., the two nudged ensembles and the late free ensemble), one can only expect them to capture the component of
tropospheric anomalies arising from stratospheric variability, not the component arising from tropospheric internal variability. In fact, the magnitude of mid-latitude surface anomalies forecasted by both nudged ensembles and the late free ensemble does agree well with the magnitude of the predictable component in ERA5 (Table 3), which is about one-fourth of the full anomalies in ERA5. These results further support that stratospheric variability accounts for about a quarter of the mid-latitude surface anomalies after the 2018 SSW.

The fractions are somewhat different with regard to the subpolar-polar surface anomalies (Table 4). For example, the predictable component of $\Delta\mathrm{SLP}_{\mathrm{Arctic}}$ is estimated to be 1.9 hPa, which is about half of the full anomaly in ERA5 (3.7 hPa). In this case, the full anomaly in ERA5 (3.7 hPa) is comparable to the forecasted anomaly in the late nudged ensemble (4.3 hPa) and the late free ensemble (4.0 hPa). The predictable component of $\Delta\mathrm{PRCP}_{\mathrm{Scandinavia}}$ is estimated to be -0.1 mm/day, which is about one-tenth of the full anomaly in ERA5 (-1.0 mm/day). In this case, the full anomaly in ERA5 (-1.0 mm/day) is in the
range of three to five times the forecasted anomaly in the early nudged ensemble (-0.3 mm/day), the late nudged ensemble (-0.2 mm/day), and the late free ensemble (-0.3 mm/day). It is unclear why the forecasted anomalies in subpolar-polar regions are stronger than the corresponding predictable component in ERA5. One potential way to address this question is to identify ma-

jor drivers of those subpolar-polar region anomalies and then evaluate to what extent those drivers are represented in SNAPSI S2S models, which is beyond the scope of the present work.

A notable feature in Tables 3-4 is the large inter-model spread in $\Delta T_{100}$ from the free ensembles. For example, for the late free ensemble, while the multi-model ensemble mean $\Delta T_{100}$ (4.6 K) agrees well with the $\Delta T_{100}$ in ERA5 (4.8 K), the inter-model spread of $\Delta T_{100}$ is as large as 2.6 K. This is because while each model's late free ensemble (Fig. 7a, diamond) predicts a warm anomaly in the stratosphere, the magnitude of the forecasted warm anomaly varies largely from model to model. For instance, $\Delta T_{100}$ from GLOBO's late free ensemble is smaller than 1.0 K (Fig. 7a, yellow diamond) whereas $\Delta T_{100}$ from CNRM-CM61's late free ensemble is larger than 7.0 K (Fig. 7a, orange diamond). Amongst the eight models, IFS is the only model whose late free ensemble provides an accurate forecast of $\Delta T_{100}$ (Fig. 7a, dark red diamond). In general, there is a linear relationship between the magnitude of the stratospheric anomaly and the magnitude of the tropospheric anomaly, consistent with previous work (White et al., 2020, 2022). That is, for those models with a relatively strong (weak) stratospheric anomaly, the corresponding tropospheric anomaly is also strong (weak). For example, for GLOBO whose late free ensemble predicts the weakest $\Delta T_{100}$, the forecasted $\Delta SLP_{Atlantic}$ and $\Delta PRCP_{Iberia}$ are also at the lower end among the eight models (Fig. 7a-b, yellow diamond); for CNRM-CM61 whose late free ensemble predicts the strongest $\Delta T_{100}$, the forecasted $\Delta SLP_{Atlantic}$ and $\Delta PRCP_{Iberia}$ are also at the higher end among the eight models (Fig. 7a-b, orange diamond); for IFS whose late free ensemble successfully captures $\Delta T_{100}$, it also captures the predictable component of $\Delta SLP_{Atlantic}$ and $\Delta PRCP_{Iberia}$ (Fig. 7a-b, dark red diamond vs. the red dot). These results indicate a close inter-model correspondence between the magnitude of the stratospheric anomaly and the magnitude of the tropospheric anomaly. An even closer inter-model correspondence exists between the magnitude of $\Delta SLP_{Atlantic}$ and the magnitude of $\Delta PRCP_{Iberia}$ (Fig. 7c). As can be seen, the eight data points indicating the late free ensembles (diamond) lie well along the best-fit line to the 46 historical SSW events in ERA5 (indicated by the solid black line Fig. 7c). That is, for those models with a stronger cyclonic $\Delta SLP_{Atlantic}$, the corresponding $\Delta PRCP_{Iberia}$ also tends to be larger. This is also true for historical SSW events in ERA, for which the correlation coefficient between $\Delta SLP_{Atlantic}$ and $\Delta PRCP_{Iberia}$ across the 46 historical SSW events is -0.85 (significant at the $p < 0.01$ level by a Student's $t$-test). That is, for those SSW events with a stronger cyclonic $\Delta SLP_{Atlantic}$, a stronger $\Delta PRCP_{Iberia}$ can be expected.

The above results reveal a close correspondence between $\Delta SLP_{Atlantic}$ and $\Delta PRCP_{Iberia}$, indicating that $\Delta SLP_{Atlantic}$ may be used as a good predictor of $\Delta PRCP_{Iberia}$. This conclusion can be further evidenced by the result of CMIP6 models (Fig. S16, given that the 28 data points indicating the 28 CMIP6 models (gray dots in panel c of Fig. S16) follow the best-fit line closely and display a high degree of overlap with the SNAPSI models' late free ensembles (diamond in panel c of Fig. S16). While SNAPSI and CMIP6 models both capture the close correspondence between $\Delta SLP_{Atlantic}$ and $\Delta PRCP_{Iberia}$, SNAPSI models appear to capture the downward coupling from the stratosphere to the mid-latitude North Atlantic troposphere better than CMIP6 models (panels a-b of Fig. S16). In particular, the eight data points indicating SNAPSI models' late free ensembles are fairly evenly distributed around the best-fit line, with a mix of data points positioned above and below the line (diamond in panels a-b of Fig. S16); as a result, the ensemble mean of all eight SNAPSI models agrees well with the predictable component of $\Delta SLP_{Atlantic}$ and $\Delta PRCP_{Iberia}$ (Table 3). In contrast, most data points indicating CMIP6 models fall on the same side of the best-fit line (gray dots in panels a-b of Fig. S16), with their y-values (indicating $\Delta SLP_{Atlantic}$ and $\Delta PRCP_{Iberia}$ in panels a and

b of Fig. S16, respectively) being consistently weaker in magnitude than the corresponding values predicted by the best-fit line. This is consistent with Dai et al. (2024) who find that the magnitude of the mid-latitude North Atlantic large-scale circulation anomalies and Iberian precipitation anomalies associated with SSWs tends to be underestimated in most CMIP6 models (see their Fig. 11 and Fig. S10).

While SNAPSI models perform well in capturing the mid-latitude surface anomalies following the 2018 SSW (Table 3 and Fig. 7), they perform less effectively in capturing the subpolar-polar surface anomalies. For example, as the only model whose late free ensemble successfully captures $\Delta T_{100}$ (Fig. 7a, dark red diamond), IFS's late free ensemble provides an accurate forecast of the predictable component of both $\Delta SLP_{Atlantic}$ and $\Delta PRCP_{Iberia}$ (Fig. 7a-b, dark red diamond). However, in terms of $\Delta SLP_{Arctic}$, the forecasted anomaly by IFS's late free ensemble (Fig. 8a, dark red diamond) is even higher than the full value of $\Delta SLP_{Arctic}$ in ERA5 (Fig. 8a, black dot) which is about twice the predictable component of $\Delta SLP_{Arctic}$ (Fig. 8a, red dot). In fact, the forecasted $\Delta SLP_{Arctic}$ of most SNAPSI models' late free ensemble is larger than the full value of $\Delta SLP_{Arctic}$ in ERA5 (Fig. 8a, diamonds vs. black dot). As a result, the ensemble mean of all eight SNAPSI models' late free ensembles is about twice the predictable component of $\Delta SLP_{Arctic}$ (Table 4). Unlike the eight SNAPSI models, the 28 data points indicating the 28 CMIP6 models (gray dots in panel a of Fig. S17) follow the best-fit line closely, indicating a general agreement between $\Delta SLP_{Arctic}$ of CMIP6 models and the corresponding values predicted by the best-fit line. This is consistent with Dai et al. (2024) who find that the magnitude of the Arctic SLP response to SSWs is well represented in CMIP6 models (see their Fig. 5). These results suggest that CMIP6 models better capture the downward coupling from the stratosphere to the Arctic surface than SNAPSI models.

The above results suggest that SNAPSI models outperform CMIP6 models in capturing the downward coupling from the stratosphere to the mid-latitude North Atlantic, while CMIP6 models outperform SNAPSI models in capturing the downward coupling from the stratosphere to the Arctic surface. The reasons for the different performances between the two sets of models remain unclear and merit further investigation. Moreover, we acknowledge that the comparison between the SNAPSI and CMIP6 models lacks equivalence due to fundamental differences in their design. For example, SNAPSI models rely largely on accurate initialization whereas CMIP6 models focus more on boundary forcings rather than starting with real-time observations. Here, comparing the SNAPSI and CMIP6 models is not about replacing one with the other, but about finding opportunities to improve both sets of models by identifying what aspects of stratosphere-troposphere coupling are better represented in one model set versus the other. That being said, Garfinkel et al. (2025) recently assessed downward coupling strength from the lower stratosphere to the near-surface over the Arctic in 22 S2S models (including most of the SNAPSI models) and found that most overestimate this coupling.

Similar to the late free ensemble, the early free ensemble also exhibits a large inter-model spread (Fig. 7a-c and Fig. 8a-c, pentagram), except that the stratospheric and tropospheric anomalies in the early free ensemble have the opposite sign from those in the late free ensemble. In particular, for the early free ensemble, every model predicts a stratosphere cooling (Fig. 7a and Fig. 8a, pentagram). In association with the stratospheric cooling, most models (except CNRM-CM61) predict an anticyclonic SLP anomaly over the Atlantic (Fig. 7a, pentagram) and dry conditions in Iberia (Fig. 7b, pentagram), as well as a cyclonic SLP anomaly over the Arctic (Fig. 8a, pentagram) and wet conditions in Scandinavia (Fig. 8b, pentagram). Again,

there is a general correspondence between the stratospheric anomaly and the tropospheric anomaly: for those models with a stronger cooling in the stratosphere, there tends to be a stronger anticyclonic SLP over the Atlantic and drier conditions over Iberia (Fig. 7a-b, pentagram), as well as a stronger cyclonic SLP over the Arctic and wetter conditions over Scandinavia (Fig. 8a-b, pentagram).

Compared to the free ensembles, the nudged ensembles have a much narrower inter-model spread in the lower stratosphere (Table 3 and Table 4). This is because almost every single model's nudged ensemble successfully captures the observed warming in the stratosphere (hexagrams and squares within the gray box in Fig. 7a-b and Fig. 8a-b), due to the nudging technique applied to the stratosphere. Even so, some small spread in $\Delta T_{100}$ still occurs because 100 hPa is below the region of full nudging. The narrow inter-model spread in the stratosphere further constrains the inter-model spread in the troposphere: the inter-model spread of tropospheric anomalies from the nudged ensemble is always smaller than those in the corresponding free ensembles (Table 3 and Table 4). As a result, the data points for the nudged ensembles are confined within a rather narrow domain (gray box in Fig. 7a-c and Fig. 8a-c). We, therefore, zoom in on the gray box to show more details of the nudged ensembles (Fig. 7d-f and Fig. 8d-f). A notable feature is that even for the nudged ensembles, there is a close correspondence between $\Delta SLP_{Atlantic}$ and $\Delta PRCP_{Iberia}$, with a more cyclonic SLP anomaly over the Atlantic corresponding to wetter conditions in Iberia (Fig. 7f). Again, this result indicates that $\Delta SLP_{Atlantic}$ may be used as a good predictor of $\Delta PRCP_{Iberia}$, although the underlying mechanisms remain an open question.

In summary, the 2018 SSW event was known for its canonical and strong tropospheric impacts, especially the rainfall extremes over the Iberian Peninsula. Here, our results show that stratospheric variability only accounts for about one-fourth of the observed Iberian precipitation signal. As a result, only a quarter of the observed Iberian precipitation signal can be predicted by S2S models, even for those ensembles that provide a 'perfect' forecast of the sudden warming in the stratosphere.

**Table 4.** As in Table 3, but for tropospheric anomalies in the subpolar-polar region. Here, $\Delta SLP_{Arctic}$ is the area-weighted mean SLP anomaly within the entire polar cap region that extends from $60°N$ towards the North Pole ($90°N$), and spans the entire range of longitudes from $0°$ to $360°$. $\Delta PRCP_{Scandinavia}$ is the area-weighted mean precipitation anomaly within the blue box in Fig. 3.

| 12-Feb-2018 SSW | Lag [1,25] | $\Delta T_{100}$ (K) | $\Delta SLP_{Arctic}$ (hPa) | $\Delta PRCP_{Scandinavia}$ (mm/day) |
|---|---|---|---|---|
| ERA5 | full | 4.8 | 3.7 | -1.0 |
| | predictable | | 1.9±0.8 | -0.1±0.2 |
| s20180125 | nudged | 5.5±0.3 | 2.8±0.6 | -0.3±0.1 |
| | free | -5.7±3.7 | -2.6±1.9 | 0.2±0.3 |
| s20180208 | nudged | 5.8±0.4 | 4.3±1.5 | -0.2±0.1 |
| | free | 4.6±2.6 | 4.0±3.2 | -0.3±0.2 |

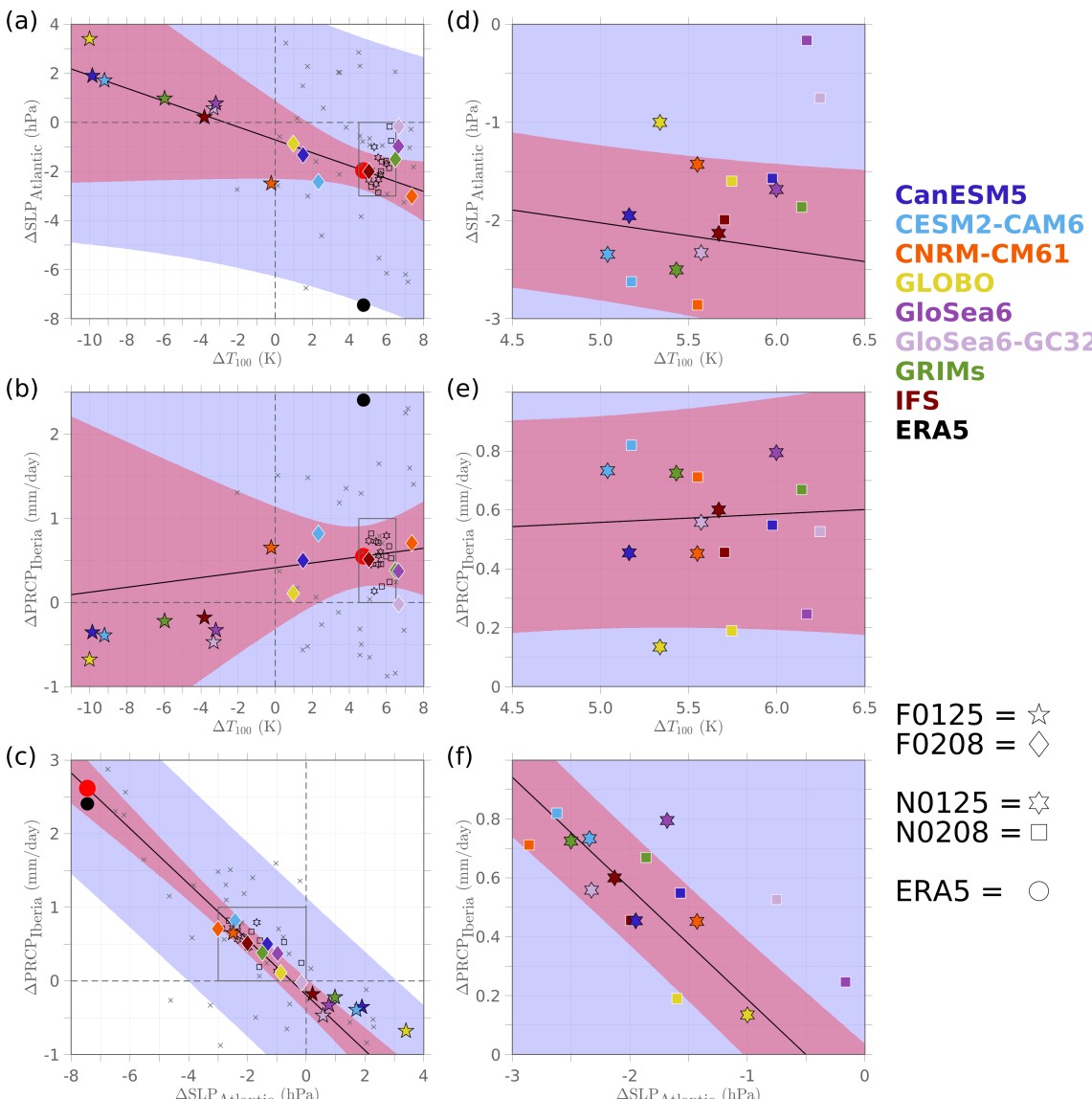

**Figure 7.** (Left) Scatter plot for values of (a) $\Delta\mathrm{SLP}_{\mathrm{Atlantic}}$ ($y$-axis) vs. $\Delta T_{100}$ ($x$-axis), (b) $\Delta\mathrm{PRCP}_{\mathrm{Iberia}}$ ($y$-axis) vs. $\Delta T_{100}$ ($x$-axis), and (c) $\Delta\mathrm{PRCP}_{\mathrm{Iberia}}$ ($y$-axis) vs. $\Delta\mathrm{SLP}_{\mathrm{Atlantic}}$ ($x$-axis). Each gray cross represents a historical SSW event in ERA5. A least-squares best fit to these gray crosses is shown as the solid black line, with the red and blue shadings indicating the 5th-95th percentile confidence interval and prediction interval, respectively. For emphasis, the 2018 SSW event from ERA5 is represented by a black dot; sitting on the best-fit line, the red dot denotes the predictable component of the 2018 SSW event. The data points indicating the eight models and four sets of forecasts are shown by corresponding shapes and colors (see the color and marker schemes to the right of the figure). F0125: early free ensemble; F0208: late free ensemble; N0125: early nudged ensemble; N0208: late nudged ensemble. The gray box encloses the data points for the nudged ensembles. For both $x$-axis and $y$-axis, dashed black lines are plotted through the zero value. (Right) As in the left panels, but only shows data points for the nudged ensembles, which are enclosed by the gray box shown in the left panels.

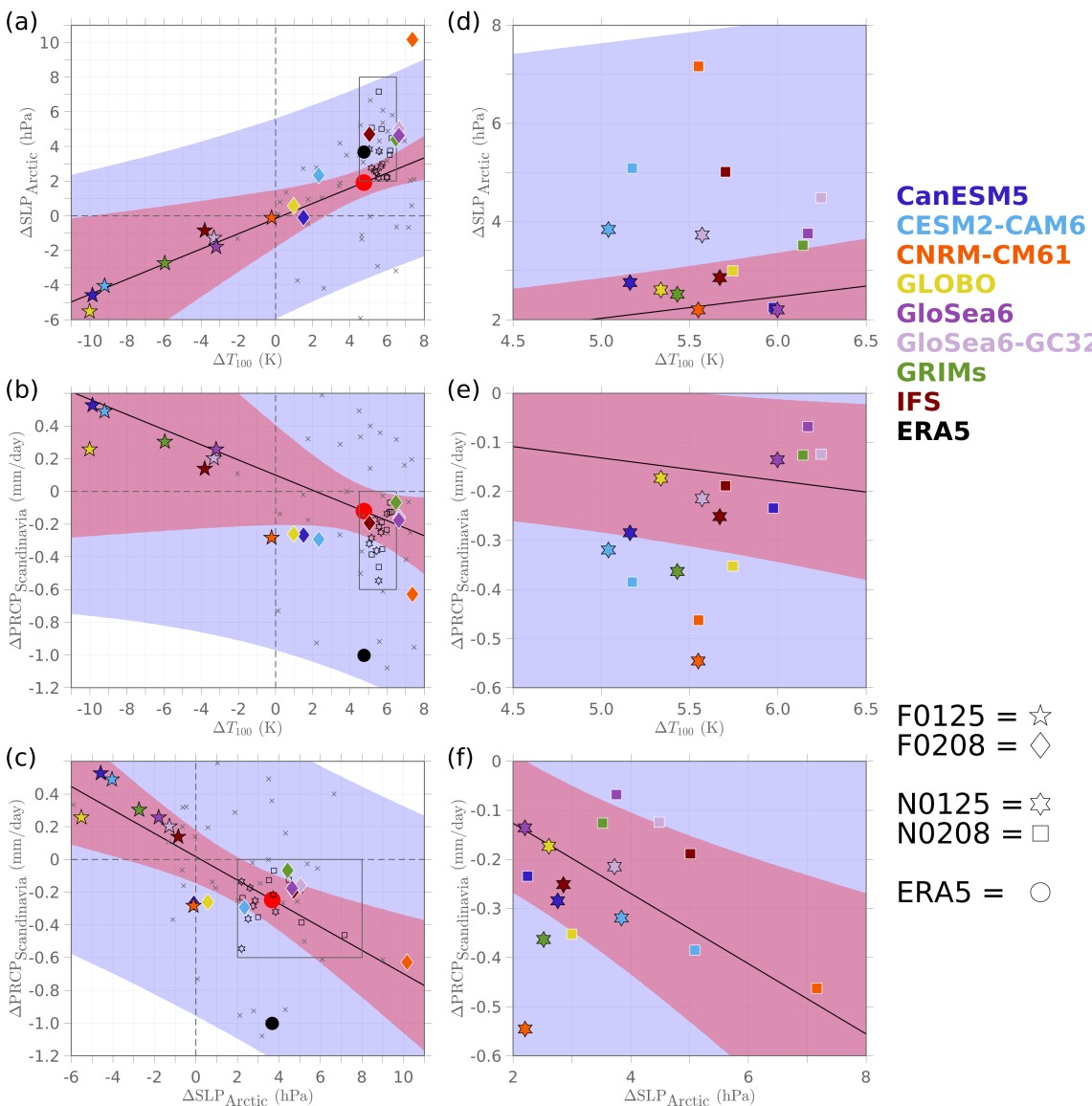

**Figure 8.** Same as Fig. 7, but for scatter plot for values of (a,d) $\Delta SLP_{Arctic}$ ($y$-axis) vs. $\Delta T_{100}$ ($x$-axis), (b,e) $\Delta PRCP_{Scandinavia}$ ($y$-axis) vs. $\Delta T_{100}$ ($x$-axis), and (c,f) $\Delta PRCP_{Scandinavia}$ ($y$-axis) vs. $\Delta SLP_{Arctic}$ ($x$-axis).

## 7   Attribution of the Iberian rainfall extreme to the 2018 SSW

According to the above results, while the 2018 SSW event is followed by record-breaking rainfall in the Iberian Peninsula (Ayarzaguena et al., 2018), only about one-fourth of the signal arises from stratospheric variability. In this case, can one still say it is the 2018 SSW event that brought extreme rainfall to Iberia? To address this question, we now examine whether the presence of the 2018 SSW increased the likelihood of Iberian rainfall extremes.

For this purpose, for each set of the ensembles, a histogram of the Iberian precipitation from all 400 members is plotted (bars in Fig. 9a-c and e-g). On top of this, a kernel distribution is estimated (curves in Fig. 9a-c and e-g), which is used to calculate the likelihood of Iberian rainfall extremes in each ensemble. Here, an Iberian rainfall extreme is defined when the Iberian precipitation is comparable to or even stronger than the one observed after the 2018 SSW (indicated by the black vertical line). For the ensemble forecasts initialized at 25 January 2018, the likelihood of Iberian rainfall extremes is $\sim 4\%$ in the control ensemble where there is no SSW in the stratosphere (Fig. 9a); the likelihood rises to $\sim 8\%$ in the nudged ensemble where the 2018 SSW is imposed in the stratosphere (Fig. 9b). The percentage difference in risk between the early nudged and control ensembles has a 5th–95th percentile uncertainty range of [2.6%, 7.1%], as determined through bootstrapping (random sampling with replacement). This range does not include zero, indicating a statistically significant difference. The almost doubled likelihood in the nudged ensemble indicates that Iberian rainfall extremes are more likely to occur in the presence of the 2018 SSW. This is also true for the ensemble forecasts initialized at 8 February 2018, for which the likelihood of Iberian rainfall extremes doubles in the nudged ensemble ($\sim 6\%$; Fig. 9f) relative to the control ensemble ($\sim 3\%$; Fig. 9e). Again, such an increase (from $\sim 3\%$ to $\sim 6\%$) is statistically significant because the percentage difference in risk between the late nudged and control ensembles has a 5th–95th percentile uncertainty range of [1.5%, 5.7%], which does not encompass zero. These results evidence the non-negligible contribution of the 2018 SSW in driving the Iberian rainfall extreme.

One may note that for the ensemble forecasts initialized on 25 January 2018, the free ensemble also shows a somewhat higher likelihood of Iberian rainfall extremes ($\sim 5\%$; Fig. 9c) than the control ensemble ($\sim 4\%$; Fig. 9a). However, such an increase is insignificant because the percentage difference in risk between the early free and control ensembles has a 5th-95th percentile uncertainty range of [-0.5%, 3.7%], which encompasses zero. This is consistent with the fact that the early free ensemble, on average, captures a cooling in the stratosphere (Fig. 9d, vertical blue line), and thus a statistically significant increase in the likelihood of Iberian rainfall extreme is not expected. By contrast, for the late free ensemble that successfully captures the stratospheric warming (Fig. 9h, vertical blue line), the likelihood of Iberian rainfall extremes ($\sim 5\%$; Fig. 9g) is significantly higher than the late control ensemble ($\sim 3\%$; Fig. 9a). This is indicated by the fact that the percentage difference in risk between the late free and control ensembles has a 5th–95th percentile uncertainty range of [0.2%, 3.8%], which does not include zero.

In summary, the record-breaking Iberian rainfall following the 2018 SSW event is more likely to occur in the presence of sudden warming in the stratosphere, indicating that the 2018 SSW event does have the potential to bring rainfall extremes to Iberia.

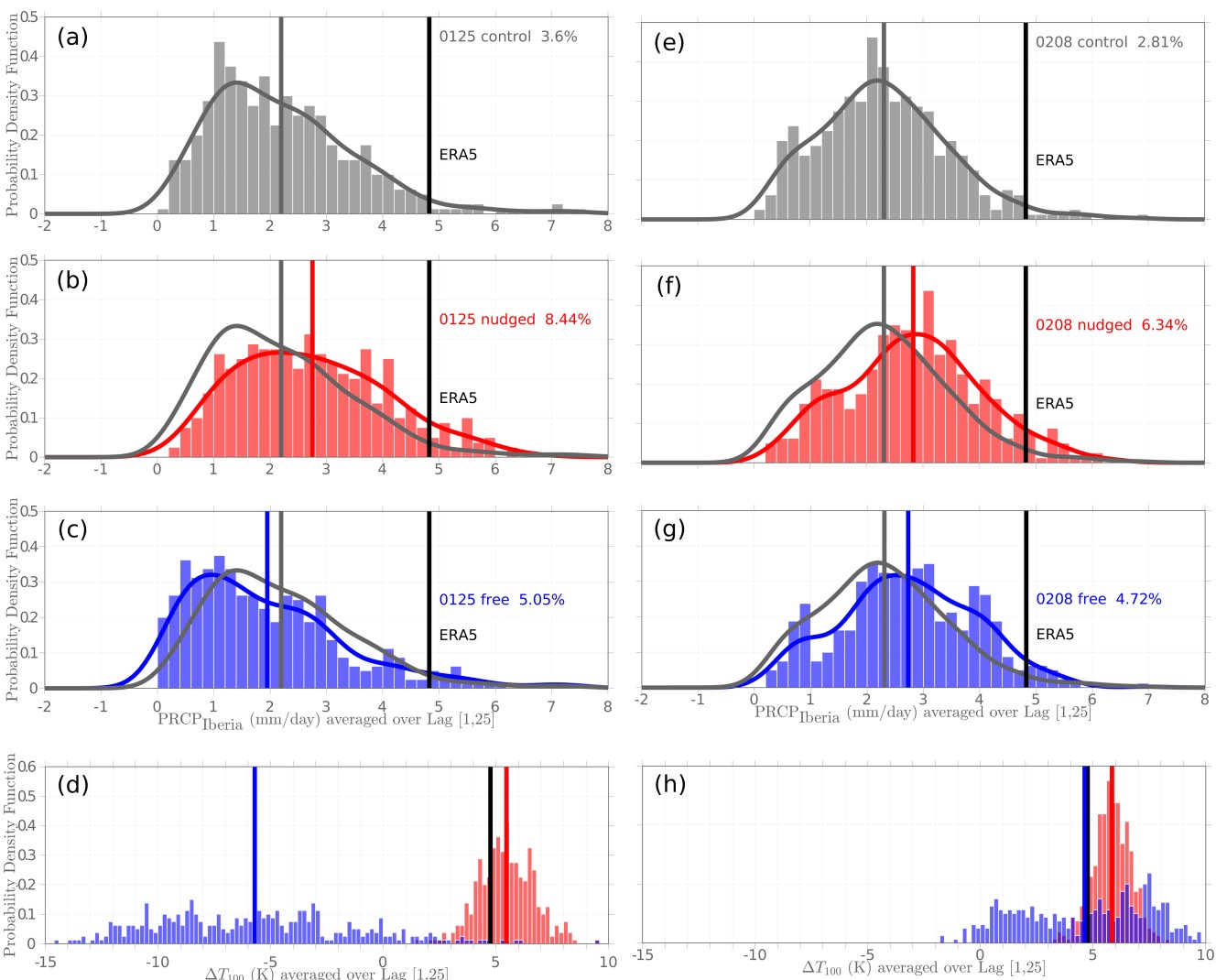

**Figure 9.** (Left) Histogram of the Iberian precipitation averaged over lag days [1,25] relative to the SSW onset date (bars) and the corresponding kernel distribution (curves) for the early (a) control, (b) nudged, and (c) free forecast ensembles. For each set of forecast ensembles, all 400 members (50 members per model; eight models in total) are used to generate the histogram and estimate the kernel distribution. The multi-model ensemble mean values are indicated by the color vertical lines, with gray, red, and blue lines corresponding to the control, nudged, and free ensemble, respectively. The value from ERA5 is indicated by the black vertical line. The percentage numbers shown on each panel are the fraction of ensemble members exceeding ERA5. (d) Histogram of $\Delta T_{100}$ from the 400 members in the early free (blue) and nudged (red) ensembles, with the blue and red vertical lines indicating the multi-model ensemble mean values. The value of $\Delta T_{100}$ from ERA5 is indicated by the black vertical line. (Right) As in the left panels, but for the late ensemble forecasts.

## 8 Understanding the early occurrence of Iberian rainfall in SNAPSI forecasts

Regarding the Iberian rainfall extreme following the 2018 SSW event, while the observed Iberian rainfall did not occur until $\sim 15$ days after the onset of the 2018 SSW (Fig. 2f), the forecasted Iberian rainfall occurs almost right after the onset of the 2018 SSW (Fig. 4f). The early occurrence of forecasted Iberian rainfall can be understood from the early occurrence of the large-scale atmospheric circulation anomaly forecasted by SNAPSI models (Fig. 10).

In the reanalysis (Fig. 10, left panels), the negative NAO-like SLP anomaly, especially the cyclonic SLP anomaly over the Atlantic, did not appear until days [11,15] after the onset of the 2018 SSW (Fig. 10c). This is consistent with Ayarzaguena et al. (2018) and González-Alemán et al. (2022) which shows that a negative NAO pattern settled in at the end of February 2018. This also explains why the Iberian rainfall did not appear until $\sim 15$ days after the onset of the 2018 SSW, given that the cyclonic SLP anomaly over the Atlantic is a pretty good predictor of the Iberian rainfall anomaly (Fig. 7c). Unlike the reanalysis, in the late free ensemble that captures the sudden warming in the stratosphere, the negative NAO-like SLP anomaly, including the cyclonic SLP anomaly over the Atlantic, appears right after the onset of the 2018 SSW (Fig. 10, right panels). As a result, the forecasted Iberian rainfall anomaly also occurs right after the onset of the 2018 SSW. This is also true for both nudged ensembles, in which the negative NAO-like SLP anomaly appears right after the onset of the 2018 SSW (Fig. S18).

In summary, the surface impact of SSW occurs sooner in S2S models than in the reanalysis, suggesting that the timing of the transition to a negative NAO that occurred at the end of February 2018 was not predictably connected to the stratospheric anomalies, at least on subseasonal timescales.

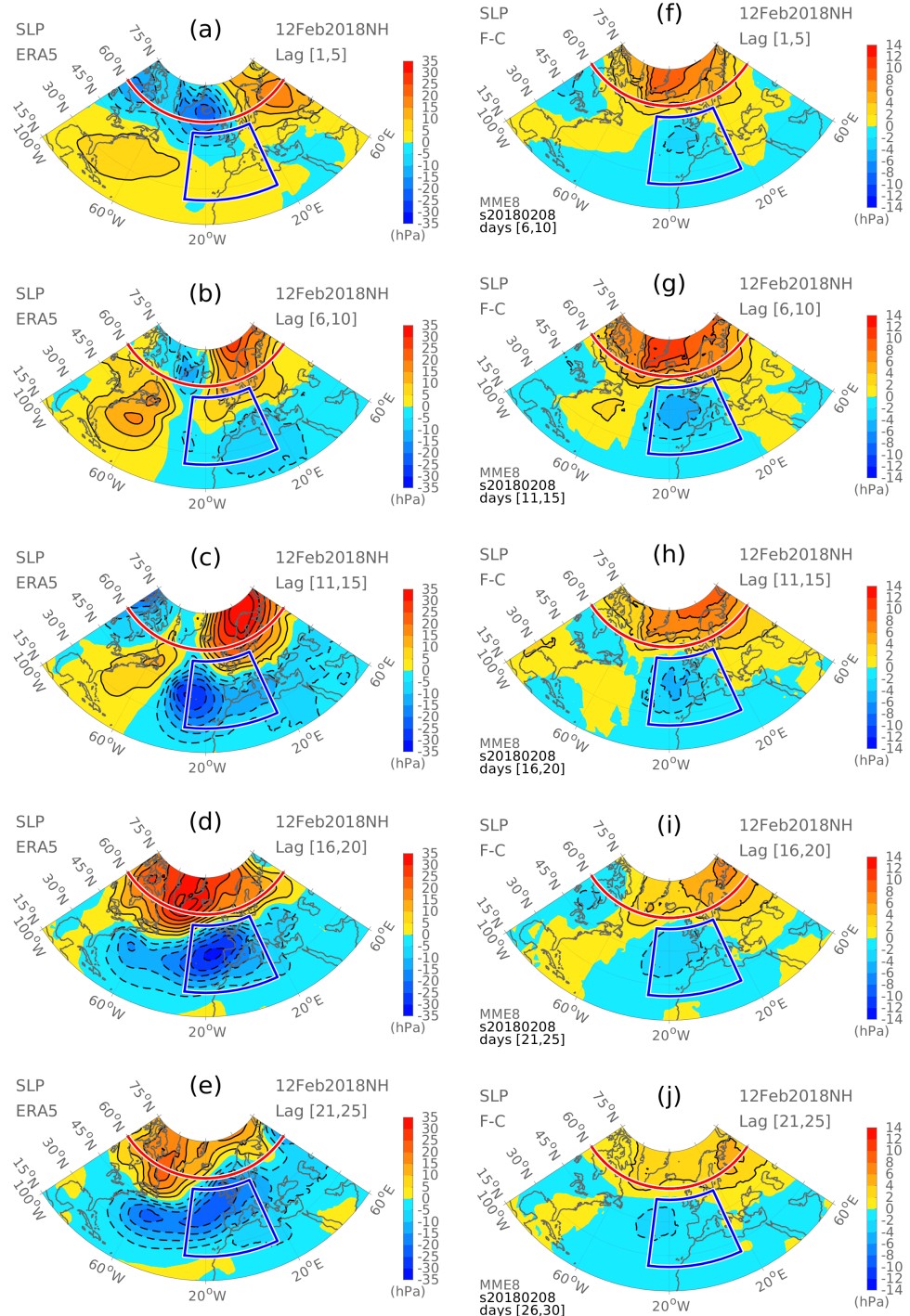

**Figure 10.** (Left) SLP anomalies from ERA5, averaged over lag days [1,5], [6,10], [11,15], [16,20], and [21,25] relative to the SSW onset date. (Right) As in the left panels, but for the multi-model ensemble mean of the late free ensembles.

## 9 Discussion

Till now, the majority of the analysis focuses on ensemble means, in which some specific characteristics of the 2018 SSW are

470 not present. For example, the ensemble mean $\Delta PRCP_{Iberia}$ is only about one-fourth as strong as the observed $\Delta PRCP_{Iberia}$ (Table 3). The timing of the ensemble mean vs. observed shift to strong, persistent negative NAO also exhibits an apparent discrepancy, with the former occurring almost right after the onset of the SSW, while the latter occurred approximately 15 days after the onset of SSW (Fig. 10). In this section, we will examine the ensemble spread of individual SNAPSI models to see if some members can capture the magnitude and timing of the observed tropospheric anomalies after the 2018 SSW. This also

enables us to find out whether certain SNAPSI models are systematically biased in representing these characteristics of the 2018 SSW.

We start with the Iberian precipitation anomaly because it is the focus of our study. For each set of forecast ensembles, we show the most extreme data value (among the 50 ensemble members) (blue whiskers in Fig. 11). As can be seen, for seven of the eight SNAPSI models analyzed in this study, some ensemble members predict Iberian precipitation anomalies comparable

to reanalysis (blue whiskers encompass ERA5). The only exception is GLOBO, for which none of the ensemble members capture the observed magnitude of the Iberian precipitation anomaly, regardless of the initialization dates or the application of the nudging. These results indicate that GLOBO is the only model that systematically underestimates the magnitude of the Iberian precipitation anomaly.

Note that GLOBO is also the model that exhibits the smallest ensemble spread compared to all other models (black whiskers

in Fig. 11). Given that the ensemble spread includes tropospheric variability, the very narrow ensemble spread of GLOBO suggests that the model's bias (namely, an underestimation of the magnitude of the Iberian precipitation anomaly) might arise from an insufficient representation of tropospheric variability. To examine whether this is the case, for each set of forecast ensembles, we show the multi-member average (color markers in Fig. 12) and the 1.65 standard deviations around the multi-member average (black whiskers in Fig. 12). We then compare the multi-member average to the predictable component of

$\Delta PRCP_{Iberia}$ from ERA5 (solid black line in Fig. 12), and compare the 1.65 standard deviations to the 90% prediction intervals from ERA5 (dashed black line in Fig. 12). Here, 1.65 standard deviations is compared to the 90% prediction intervals because mean $\pm$ 1.65 standard deviations covers the middle 90% if we assume a normal distribution. As expected, GLOBO's ensemble spread is notably smaller than the 90% prediction intervals from ERA5, evidencing an insufficient representation of tropospheric variability in GLOBO.

We then examine the evolution of the Atlantic SLP anomaly in individual ensemble members because the Atlantic SLP anomaly is a good predictor of the Iberian precipitation anomaly (Fig. 7c). To this end, for the late free ensemble of every SNAPSI model, we show the daily Atlantic SLP anomaly over days [1,25] from each of the 50 ensemble members (color lines in Fig. 13). Here, the result of the late free ensemble is shown as a supplement to Fig. 10, of which the right panels show the multi-model ensemble mean of the late free ensemble. It turns out that, for most SNAPSI models, individual ensemble

members envelope the observed evolution of the Atlantic SLP anomaly (black lines in Fig. 13) throughout most of the period of interest. Hence, there is no evidence that these SNAPSI models are systematically biased in capturing the timing of shift to

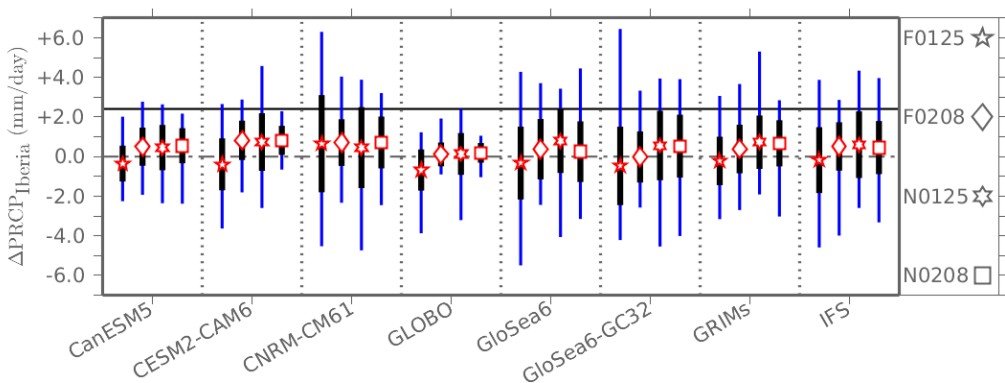

**Figure 11.** $\Delta$PRCP$_{\text{Iberia}}$ by individual SNAPSI models. For each ensemble, the color marker is the multi-member average, the black whiskers correspond to 1 standard deviation around the multi-member average, and the blue whiskers extend to the most extreme data value. F0125: early free ensemble; F0208: late free ensemble; N0125: early nudged ensemble; N0208: late nudged ensemble. The solid black horizontal line indicates $\Delta$PRCP$_{\text{Iberia}}$ from ERA5.

a negative NAO. Again, an exception is GLOBO (Fig. 13b), for which the observed response falls outside of the envelope of individual ensemble members throughout most of the period of interest.

These results suggest that for most SNAPSI models (except GLOBO), several individual ensemble members predict Iberian precipitation anomalies comparable to reanalysis. The observed evolution of the Atlantic SLP anomaly is also enveloped by individual ensemble members in those models, indicating that most SNAPSI models are not systematically biased in capturing the magnitude and timing of surface anomalies following the 2018 SSW.

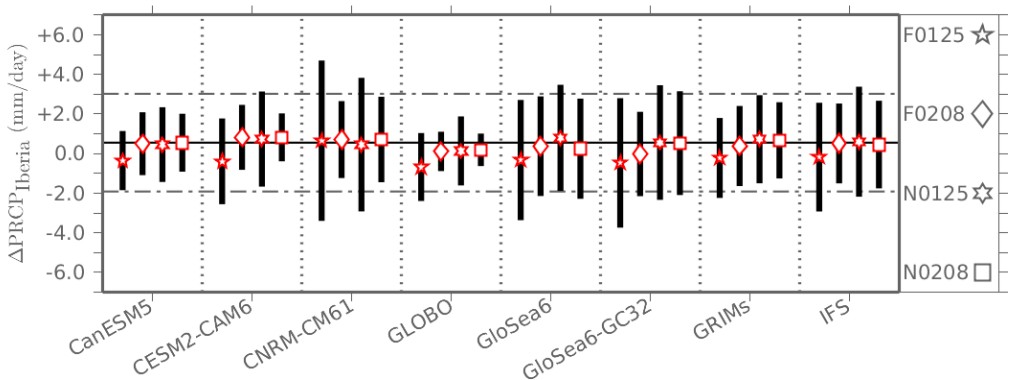

**Figure 12.** $\Delta$PRCP$_{\text{Iberia}}$ by individual SNAPSI models. For each ensemble, the color marker is the multi-member average, and the black whiskers correspond to 1.65 standard deviations around the multi-member average. F0125: early free ensemble; F0208: late free ensemble; N0125: early nudged ensemble; N0208: late nudged ensemble. The solid black horizontal line indicates the predictable component of $\Delta$PRCP$_{\text{Iberia}}$ from ERA5, and the dashed black lines indicate the 90% prediction intervals (the predictable component and the prediction intervals are the same with those indicated by the red dot and the corresponding blue shadings in Fig. 7b, which are obtained from a linear regression model built between $\Delta$PRCP$_{\text{Iberia}}$ and $\Delta T_{100}$ using historical SSW events from ERA5).

## 10 Conclusions

The 2018 SSW event, which occurred on 12 February 2018, was followed by the canonical negative NAO-like weather regime, which brought dry spells to Scandinavia and record-breaking rainfall to the Iberian Peninsula. Identified in global climate models and reanalyses, the precipitation signal following the 2018 SSW has never been confirmed in observations yet. In this study, we first identify the precipitation signal following the 2018 SSW using satellite and in situ station observations of precipitation. It turns out that both the Scandinavian dry spell and the Iberian record-breaking rainfall are identifiable in observations, of which the former appeared almost right after the onset of the 2018 SSW whereas the latter did not appear until 15 days after the onset of the 2018 SSW. As the first study to identify stratospheric signals linked to SSW in observational precipitation datasets, we find a close agreement between observational and ERA5 precipitation datasets in capturing the precipitation anomalies following the 2018 SSW, which sharpens the realism and strengthens the confidence in our analyses.

After confirming the precipitation signal in observations, we investigate its predictability in state-of-the-art S2S models using a new multi-model database of S2S forecasts generated by the SNAPSI project. We find that the observed 'Wet Iberia and Dry Scandinavia' pattern is captured by the free and nudged ensemble initialized at 8 February 2018, both of which successfully captures the sudden warming in the stratosphere. By contrast, a stratospheric cooling is forecasted by the free ensemble initialized at 25 January 2018; in association, a 'Dry Iberia and Wet Scandinavia' precipitation signal is forecasted. Initialized at the same date with the early free ensemble, the early nudged ensemble captures the sudden warming in the stratosphere (by design) as well as the 'Wet Iberia and Dry Scandinavia' pattern. The distinct behavior between the free and

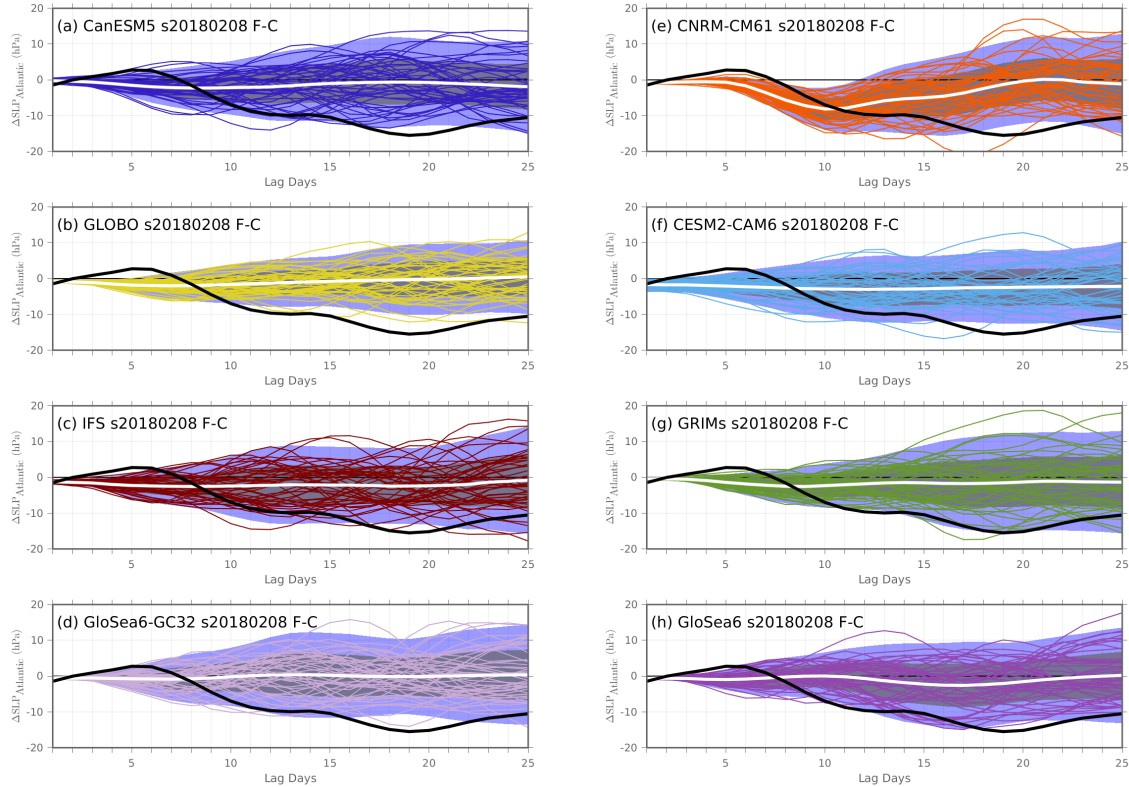

**Figure 13.** Time series of the Atlantic SLP anomaly over days [1,25] after the 2018 SSW from ERA5 (thick black curve) and the late free ensemble (thin color curves and thick white curve). For the late free ensemble, the thin color curves indicate 50 ensemble members for each model, and then the thick white curve indicates the ensemble mean of 50 members. Grey and blue shading correspond to 1 and 2 standard deviations around the multi-member average. Here, a 5-day moving average is applied to reduce high-frequency variations.

and nudged ensembles initialized at 25 January 2018 highlights the importance of getting a successful stratospheric forecast, which can enhance predictability of the precipitation signal for several weeks.

While the 'Wet Iberia and Dry Scandinavia' pattern is well captured by the ensembles with a successful stratospheric forecast, the magnitude of the forecasted precipitation signal in the ensemble man is much weaker than that observed. To understand whether models are underestimating the precipitation signal, we identify a systematic component of precipitation anomalies associated with stratospheric variability using historical SSWs in ERA5. It turns out that the ensemble-mean magnitude of the forecasted Iberia precipitation anomaly agrees well with the 'predictable component' of precipitation anomalies estimated from historical SSW events in ERA5; both are about one-fourth of the full anomaly in observations. These results suggest that stratospheric variability alone accounts for a quarter of the extreme precipitation over Iberia after the 2018 SSW. This suggests that the exceptionally strong surface anomalies following the 2018 SSW are probably significantly driven by tropospheric variability. In fact, while the 2018 SSW ranks among the strongest on record in terms of the subsequent surface anomalies, the

sudden warming itself was not particularly strong (Fig. S19). The contrast between the case's moderate intensity and its disproportionately strong surface impact also points to the substantial role of tropospheric variability in driving the observed surface anomalies following the 2018 SSW. While we conclude that the models realistically represent the predictable component of the Iberian precipitation response to the SSW, there is some evidence that Arctic response is overestimated by these models, consistent with the analysis of the S2S hindcast data from many of these models in Garfinkel et al. (2025).

Since the stratospheric variability accounts for only a quarter of the extreme rainfall anomaly over Iberia, can one still attribute the Iberian rainfall extreme to the 2018 SSW? To address this question, we calculate the likelihood of extreme Iberian rainfall in each set of forecast ensemble. Compared with the control ensemble, the nudged ensemble exhibits a doubled likelihood of Iberian rainfall extremes whose strength is comparable to or even stronger than the one observed following the 2018 SSW. The doubled likelihood of Iberian rainfall extremes in the presence of the 2018 SSW suggests that the 2018 SSW event does have the potential to bring rainfall extremes to Iberia.

The findings of this study indicate that the stratosphere represents an important source of S2S predictability for precipitation over Europe and call for consideration of stratospheric variability in hydrological prediction at S2S timescales.

*Data availability.* The IMERG data used in this study comes from https://gpm.nasa.gov/data/imerg. The GSN data used in this study comes from https://gcos.wmo.int/en/networks/atmospheric/gsn. The ERA5 data used in this study comes from https://cds.climate.copernicus.eu/datasets. The SNAPSI forecasts used in this study are archived by CEDA (https://catalogue.ceda.ac.uk/uuid/0a5a1ce22fb047749e040879efa8e9b5).

*Supplement.* The supplement related to this article is available.

*Author contributions.* Y.D. and P.H. initially conceived of and designed the study. Y.D. performed the analysis and wrote the first draft of the manuscript. P.H., A.H.B., and C.I.G. coordinated the SNAPSI runs. All authors discussed the results and edited the manuscript.

*Competing interests.* Some authors are members of the editorial board of journal Weather and Climate Dynamics.

*Acknowledgements.* YD and PH acknowledge support from NASA through the Subseasonal-to-Seasonal Hydrometeorological Prediction program under award 80NSSC22K1838. CIG is supported by ISF grant No. 3065/23 and United States-Israel Binational Science Foundation grant no. 2021714. This work used JASMIN (Lawrence et al., 2013), the UK's collaborative data analysis environment (https://www.jasmin.ac.uk).

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
