# Peer review of "Assessing stratospheric contributions to subseasonal predictions of precipitation after the 2018 SSW from SNAPSI"

_EGUsphere, 2025_

## Author Comment (AC1)

We would like to thank Reviewer 1 for the very helpful comments. Please find below our responses to all of the reviewer's comments.

Reviewer #1: Reviewer Comments

The manuscript investigates the role of the 2018 SSW in S2S prediction of the "Wet Iberia and Dry Scandinavia" precipitation pattern by using observations and a new S2S forecasts project SNAPSI. Some of the conclusions are - the early nudged ensembles, unlike the early free ensembles, successfully capture the "Wet Iberia and Dry Scandinavia" precipitation pattern, suggesting that an accurate forecast of stratospheric variability is helpful to improve S2S predictability of precipitation. In addition, the SSW related precipitation anomaly accounts for about $1/4$ of the observed precipitation anomaly, suggesting the importance of other tropospheric variability.

The manuscript covers an important topic in assessing and quantifying the contribution of SSW on the "Wet Iberia and Dry Scandinavia" pattern in 2018. It's also overall well-written. However, I have some major comments about assessing the role of tropospheric internal variability using ensemble spread.

**Major comments:**

**Comment 1:** About the role of tropospheric internal variability. While I understand that the majority of the analysis focuses on ensemble mean to extract the SSW-induced variability, I would suggest the authors examine more on the ensemble spread since the ensemble spread includes the tropospheric variability. Some of the questions that the authors could investigate include - Can some ensemble members capture the observed magnitude of the precipitation pattern? (This seems to be yes given Fig. 9 but the question is different from Section 7) What does the ensemble mean versus ensemble spread tell about the role of tropospheric variability? (The role of tropospheric variability is assessed in the paper by using the linear regression approach. Does the model simulations tell the same attribution?) What are the model-to-model differences?

*Reply:*

- "Can some ensemble members capture the observed magnitude of the precipitation pattern?"

  To address this question, we show the most extreme data value for each set of forecast ensembles (blue whiskers in Fig. 1 in this reply). As can be seen, for seven of the eight SNAPSI models analyzed in this study, several individual ensemble members predict Iberian precipitation anomalies comparable to reanalysis (blue whiskers encompass ERA5 in Fig. 1). The only exception is GLOBO, for which none of the ensemble members capture the observed magnitude of the precipitation pattern, regardless of the initialization dates or the application of the nudging technique (blue whiskers remain below ERA5).

- "What does the ensemble mean versus ensemble spread tell about the role of tropospheric variability? (The role of tropospheric variability is assessed in the paper by using the linear regression approach. Does the model simulations tell the same attribution?)"

  To address this question, for each set of forecast ensembles, we show the multi-member average (color markers in Fig. 2 in this reply) and the 1.65 standard deviations around the multi-member average (black whiskers in Fig. 2). For ERA5, the predictable component of $\Delta\text{PRCP}_{\text{Iberia}}$ (solid black line in Fig. 2) and the 90% prediction intervals (dashed black line in Fig. 2) are shown. Here, 1.65 standard deviations is compared to the 90% prediction intervals because mean $\pm$ 1.65 standard deviations covers the middle 90% if we assume a normal distribution.

  Taking IFS as an example, the ensemble mean $\Delta\text{PRCP}_{\text{Iberia}}$ from the late free ensemble and the two nudged ensembles agree well with the predictable component of $\Delta\text{PRCP}_{\text{Iberia}}$ from ERA5. Given that the ensemble mean and the predictable component both represent SSW-induced variability, the agreement between them indicates that the role of stratospheric variability is well captured by IFS. With regard to the role of tropospheric variability, it is also well captured by IFS. In particular, the ensemble spread (black whiskers in Fig. 2) in general agrees with the corresponding prediction intervals from ERA5 (horizontal dashed lines in Fig. 2). This is, in general, the case for the rest of the models.

The only exception is GLOBO, for which the ensemble spread is much smaller than the prediction intervals from ERA5.

- "What are the model-to-model differences?"

  Please see our reply to the above two questions.

Based on the above results, we have added a discussion section (the new section 9) to the revised manuscript. Please see lines 468-506 of the revised manuscript.

**Comment 2:** About 2018 SSW compared to other SSWs. I would suggest the authors add some conclusions/discussion on how 2018 SSW is compared to other SSWs. As shown in Fig. 7, 2018 SSW is not particularly strong based on T100 but the SLP_Atlantic is the strongest. This suggests that the 2018 SSW is not very different from other SSWs and the SLP_Atlantic and precipitation pattern is mostly driven by other tropospheric variability. This information is here and there in the paper but would be very useful information to include in Conclusions/Discussion.

*Reply:* To address this question, we show a boxplot of $\Delta T_{100}$, $\Delta SLP_{Atlantic}$, and $\Delta PRCP_{Iberia}$ for 46 historical SSW events from ERA5 (Fig. 3 in this reply). As can be seen, $\Delta T_{100}$ of the 2018 SSW is close to the median value, indicating the moderate strength of the sudden warming itself. By contrast, $\Delta SLP_{Atlantic}$ and $\Delta PRCP_{Iberia}$ following the 2018 SSW fall within the bottom and top 25th percentile of all historical events, respectively. The contrast between the moderate intensity of the case and the disproportionately strong surface anomaly indicates that the observed surface anomalies following the 2018 SSW are probably significantly driven by tropospheric variability.

Accordingly, we have made changes to the conclusion section in the revised manuscript. Please see lines 532-537 of the revised manuscript, where we write:

*"This suggests that the exceptionally strong surface anomalies following the 2018 SSW are probably significantly driven by tropospheric variability. In fact, while the 2018 SSW ranks among the strongest on record in terms of the subsequent surface anomalies, the sudden warming itself is not particularly strong (Fig. S19). The contrast between the case's moderate intensity and its disproportionately strong surface impact also points to the substantial role of tropospheric variability in driving the observed surface anomalies following the 2018 SSW."*

**Comment 3:** This might be minor but why all the free running models predict a stratospheric cooling?
*Reply:*

It has been pointed out by previous studies that the S2S prediction systems (from the S2S database) cannot forecast the 2018 SSW event more than 10 days in advance (Karpechko et al., 2018; Rao et al., 2020; Butler et al., 2020). This is because the 2018 SSW event was associated with primarily wavenumber-2 wave forcing that was not well predicted more than 7-10 days ahead of time (Butler et al., 2020). For example, according to Fig. 1a of Karpechko et al. (2018) (or Fig. 4 in this reply), at a lead time of 18 days (initialized from 25 January 2018), none of the S2S models from the S2S database predict an SSW (grey dots in Fig. 4). Our results show that this is also the case for SNAPSI models because none of the SNAPSI models' ensemble mean forecast from 25 January 2018 (at a lead time of 18 days) predicted an SSW.

It has also been pointed out by a previous study that, initialized at 25 January 2018, all S2S models show a positive Stratospheric Polar Vortex (SPV) error [green dots in Fig. 10b of Butler et al. (2020); or Fig. 5 in this reply]. The positive SPV error is consistent with the stratospheric cooling predicted by SNAPSI models early free ensembles. Understanding the origin of the strong vortex error is beyond the scope of this study, and here are just a couple of thoughts.

- (a) Errors in the stratosphere forecasts might have contributed to errors in the SPV forecasts.

  Butler et al. (2020) found that the SPV error (too strong polar vortex winds) correlates significantly with the QBO error (see the correlation values in the upper-right corner of Fig. 5 in this reply). This suggests that the SPV error is larger in those systems that have a larger QBO error. Hence, the SPV state can be better predicted in those systems that better predict the QBO state, highlighting the importance of reducing errors within the stratosphere (particularly in the QBO forecasts).

- (b) Errors in the troposphere (particularly in the wave-2 activity) forecasts might have contributed to errors in the SPV forecasts.

  Wu et al. (2024) compared two clusters of ensemble forecasts, one with ensemble members that successfully predict the SSW ('SSW cluster') and one that predicts a strong vortex state ('strong vortex cluster'), to understand the origin of the different predictions of the vortex strength. Both the SSW cluster and the strong vortex are initialized 16 days before the onset of the 2018 SSW event [Fig. 4a of Wu et al. (2024); or Fig. 6a in this reply], close to the initialization date of the early free ensemble forecasts analyzed in this study. It turns out that the two clusters do not differ significantly in wave-1 activity (Fig. 6b in this reply). By contrast, the two clusters differ significantly in wave-2 activity around the onset of the 2018 SSW. In particular, both clusters show an initial increase (Fig. 6c in this reply) in wave-2 activity, but the increase in the strong vortex cluster is much weaker and less persistent than that in the SSW cluster. These results suggest that the strong vortex state in the strong vortex cluster may arises from the errors in the tropospheric wave-2 activity forecasts. In particular, the amplification of tropospheric wave-2 activity in the strong vortex cluster lacks the strength and persistence required to disrupt the polar vortex state.

  In this sense, the question of why the early free ensembles predict a stratospheric cooling may be understood as follows: first of all, the early free ensembles are initialized under a strong vortex state [indicated by the positive NAM index around the orange line in Fig. 5 of Hitchcock et al. (2022); or Fig. 7 in this reply]. The strong vortex state is then maintained due to the weak and short-lived nature of the wave-2 forcing. As a consequence, a stratospheric cooling is predicted in the early free ensembles.

**References**

Butler, A. H., Lawrence, Z. D., Lee, S. H., Lillo, S. P., and Long, C. S.: Differences between the 2018 and 2019 stratospheric polar vortex split events, Quarterly Journal of the Royal Meteorological Society, 146, 3503–3521, https://doi.org/https://doi.org/10.1002/qj.3858, 2020.

Hitchcock, P., Butler, A., Charlton-Perez, A., Garfinkel, C. I., Stockdale, T., Anstey, J., Mitchell, D., Domeisen, D. I. V., Wu, T., Lu, Y., Mastrangelo, D., Malguzzi, P., Lin, H., Muncaster, R., Merryfield, B., Sigmond, M., Xiang, B., Jia, L., Hyun, Y.-K., Oh, J., Specq, D., Simpson, I. R., Richter, J. H., Barton, C., Knight, J., Lim, E.-P., and Hendon, H.: Stratospheric Nudging And Predictable Surface Impacts (SNAPSI): a protocol for investigating the role of stratospheric polar vortex disturbances in subseasonal to seasonal forecasts, Geoscientific Model Development, 15, 5073–5092, https://doi.org/10.5194/gmd-15-5073-2022, 2022.

Karpechko, A. Y., Perez, A. C., Balmaseda, M., Tyrrell, N., and Vitart, F.: Predicting sudden stratospheric warming 2018 and its climate impacts with a multi-model ensemble, Geophysical Research Letters, p. 2018GL081091, 2018.

Rao, J., Garfinkel, C. I., and White, I. P.: Predicting the Downward and Surface Influence of the February 2018 and January 2019 Sudden Stratospheric Warming Events in Subseasonal to Seasonal (S2S) Models, Journal of Geophysical Research: Atmospheres, 125, e2019JD031 919, https://doi.org/https://doi.org/10.1029/2019JD031919, e2019JD031919 2019JD031919, 2020.

Wu, R. W.-Y., Chiodo, G., Polichtchouk, I., and Domeisen, D. I. V.: Tropospheric links to uncertainty in stratospheric subseasonal predictions, Atmospheric Chemistry and Physics, 24, 12 259–12 275, https://doi.org/10.5194/acp-24-12259-2024, 2024.

[Figure]

Figure 1: $\Delta$PRCP$_{\text{Iberia}}$ by individual SNAPSI models. For each ensemble, the color marker is the multi-member average, the black whiskers correspond to 1 standard deviation around the multi-member average, and the blue whiskers extend to the most extreme data value. F0125: early free ensemble; F0208: late free ensemble; N0125: early nudged ensemble; N0208: late nudged ensemble. The solid black horizontal line indicates $\Delta$PRCP$_{\text{Iberia}}$ from ERA5.

[Figure]

Figure 2: $\Delta\mathrm{PRCP}_{\mathrm{Iberia}}$ by individual SNAPSI models. For each ensemble, the color marker is the multi-member average, and the black whiskers correspond to 1.65 standard deviations around the multi-member average. F0125: early free ensemble; F0208: late free ensemble; N0125: early nudged ensemble; N0208: late nudged ensemble. The solid black horizontal line indicates the predictable component of $\Delta\mathrm{PRCP}_{\mathrm{Iberia}}$ from ERA5, and the dashed black lines indicate the 90% prediction intervals (the predictable component and the prediction intervals are the same with those indicated by the red dot and the corresponding blue shadings in Fig. 7b in the manuscript, which are obtained from a linear regression model built between $\Delta\mathrm{PRCP}_{\mathrm{Iberia}}$ and $\Delta T_{100}$ using historical SSW events from ERA5).

[Figure]

Figure 3: $\Delta T_{100}$, $\Delta\mathrm{SLP}_{\mathrm{Atlantic}}$, and $\Delta\mathrm{PRCP}_{\mathrm{Iberia}}$ for 46 historical SSW events from ERA5. On each box, the red line is the median, the edges of the box are the 25th and 75th percentiles, and the whiskers extend to the most extreme data value. The blue circle indicates the value for the 2018 SSW event.

[Figure]

Figure 4: Predictions of the 2018 SSW by individual S2S models. Red dots indicate that an SSW is predicted within 3 days from the actually observed event. Blue dots indicate that an SSW is predicted on a different date than the actual event. Gray dots indicate that no SSW is predicted. Arrows point to forecasts used in the multimodel ensembles. Figure 1a from Karpechko et al. (2018).

[Figure]

Figure 5: Scatter plots of the averaged error in the QBO forecasts versus error in the zonal-mean zonal winds at 60°N and 10-hPa (Stratospheric Polar Vortex) for the 2018 SSW. Correlation values are calculated using the multi-model ensemble of 201 members. An asterisk indicates that the correlation is significant at the 95% confidence level. Figure 10b from Butler et al. (2020).

[Figure]

Figure 6: (a) A comparison of the zonal-mean zonal winds at 60°N and 10-hPa between strong vortex cluster (red), SSW cluster (blue), and ERA5 (black). (b and c) same as panel a but for (b) wave-1 and (c) wave-2 component of the zonal average of meridional eddy heat fluxes at 100-hPa averaged over 45-75°N, respectively. The vertical line denotes the central date of the SSW on 12 February 2018. Figure 4 from Wu et al. (2024).

[Figure]

Figure 7: NAM indices during the February 2018 boreal major warming. The vertical dashed-dotted black line indicates the date of the wind reversal at 10-hPa, 60°N. The vertical green and orange lines indicate the requested initialization dates. Figure 5 from Hitchcock et al. (2022).

---

## Author Comment (AC2)

We would like to thank Reviewer 2 for the very helpful comments. Please find below our responses to all of the reviewer's comments.

Reviewer #2: Reviewer Comments

Review: Assessing Stratospheric Contributions to Subseasonal Predictions of Precipitation after the 2018 SSW from SNAPSI

Ying Dai, Peter Hitchcock, Amy H. Butler, Chaim I. Garfinkel, and William J. M. Seviour

Summary

The manuscript presents an in-depth analysis of the role of stratospheric extreme events, particularly the February 2018 sudden stratospheric warming (SSW), in subseasonal to seasonal (S2S) predictability of precipitation patterns over Europe. Utilizing a newly developed database of S2S forecasts from the SNAPSI project, the authors systematically compare three types of forecast ensembles: a free ensemble, a nudged-to-observations ensemble, and a control ensemble. The study's focus on the 'Wet Iberia and Dry Scandinavia' precipitation signal provides valuable insights into the extent to which stratospheric variability influences predictability.

The manuscript is well-structured, presenting a coherent progression from observational evidence to model-based analysis. The methodology is robust, employing multiple model ensembles, observational datasets, and reanalysis products to assess predictability. However, some aspects require further clarification and refinement, particularly regarding the role of tropospheric internal variability, statistical robustness, and potential implications for operational forecasting. Below, I outline specific issues that should be addressed to strengthen the manuscript. Overall, this manuscript represents a valuable contribution to the field, and with the suggested revisions, it will be well-suited for publication.

**Major comments:**

**Comment 1:** The manuscript highlights the inclusion of observational datasets (IMERG and GSN) alongside reanalysis data (ERA5) to assess the precipitation signals following the 2018 SSW. While this is an important addition compared to Ayarzaguena et al. (2018) it is not clear what additional scientific value the observational analysis provides beyond serving as a verification dataset for the reanalysis. Since ERA5 is already widely used for SSW impact studies, the authors should explicitly clarify (1) what new insights are gained by including observations, (2) whether the observations provide significant deviations from the reanalysis, and (3) how these deviations, if any, impact the overall conclusions of the study. If the observations largely confirm the ERA5 findings, it would be useful to state this explicitly and discuss the implications.

*Reply:*

- (1) what new insights are gained by including observations

  While ERA5 is already widely used for SSW impact studies, its precipitation field cannot be used as a direct proxy for observations. This is because reanalysis precipitation is a model-based product that assimilates limited observational data. For example, ERA5 does not directly assimilate any rain-gauge data (Lavers et al., 2022). As a result, ERA5 precipitation is heavily influenced by the model's own physics and parameterizations, which can introduce systematic errors and regional biases that would not be present in true observational datasets. As a consequence, ERA5 precipitation might not accurately capture the true timing, intensity, or location of precipitation events. Since we focus on an extreme precipitation event in this study, it is essential to evaluate the ability of ERA5 precipitation to capture the magnitude, location, pattern, and daily variations of the observed precipitation for there to be confidence in the ERA5 precipitation.

  In this sense, a key new insight from this study is the first identification of stratospheric signals linked to SSW in observational precipitation datasets. The close agreement between observations and reanalysis sharpens the realism and strengthens the confidence in our analyses.

- (2) whether the observations provide significant deviations from the reanalysis

  As shown in Fig. 2 in the manuscript, GSN agrees closely with ERA5 (green vs. gray curves in Fig. 2 in the manuscript).

  While the anomalies in IMERG appear stronger than those in ERA5 (purple vs. gray curves in Fig. 2 in the manuscript), the discrepancies are more plausibly due to limitations in IMERG itself, given that GSN shows good agreement with ERA5.

- (3) how these deviations, if any, impact the overall conclusions of the study. If the observations largely confirm the ERA5 findings, it would be useful to state this explicitly and discuss the implications.

  The observations largely confirm the ERA5 findings, evidencing the reliability of ERA5 reanalysis precipitation for this case.

  In the revised manuscript, we have added a discussion on the scientific value to the conclusion section. Please see lines 514-516 of the revised manuscript, where we write:

  *"As the first study to identify stratospheric signals linked to SSW in observational precipitation datasets, we find a close agreement between observational and ERA5 precipitation datasets in capturing the precipitation pattern following the 2018 SSW, which sharpens the realism and strengthens the confidence in our analyses."*

**Comment 2:** The study concludes that approximately one-quarter of the observed precipitation anomaly amplitude can be attributed to stratospheric variability, with the remaining fraction likely arising from tropospheric internal variability. Also, the peaks in rainy conditions over Iberia are explained by synoptic-scale processes. This indicates an increased importance of tropospheric variability. Do certain atmospheric configurations at the tropospheric level amplify or dampen the stratospheric signal?

**Reply:** As has been pointed out in the manuscript, the Atlantic low anomaly is a pretty good predictor of the Iberian rainfall anomaly (Fig. 7c in the manuscript). In particular, a stronger Atlantic low anomaly corresponds to a larger Iberian rainfall anomaly. In this sense, an Atlantic low (high) anomaly driven by tropospheric internal variability can amplify (dampen) the stratospheric signal in the Iberian rainfall anomaly. With that said, a detailed examination of how tropospheric configurations may amplify or suppress the stratospheric signal lies outside the scope of this study, because SNAPSI is not designed to investigate such processes.

**Comment 3:** The authors argue that the doubled likelihood of Iberian rainfall extremes in the presence of the 2018 SSW suggests that the event does have the potential to bring rainfall extremes to Iberia. How is this aligned with only a quarter of the observed precipitation anomaly amplitude being attributed to stratospheric variability in general? How does 2018 SSW differ from the other major SSW events in this regard?

**Reply:**

- How is this aligned with only a quarter of the observed precipitation anomaly amplitude being attributed to stratospheric variability in general?

  The two perspectives converge on the view that other influencing factors also play a role in driving the observed Iberian precipitation anomaly. For example, according to Fig. 9 in the manuscript, even in the presence of the 2018 SSW, the likelihood of Iberian rainfall extremes remains below 9%. This relatively low likelihood suggests that the actual occurrence of Iberian rainfall extremes does not depend solely on the occurrence of SSW but depends on a combination of other influencing factors. Similarly, the fact that the 2018 SSW only accounts for a fraction of the observed precipitation anomaly amplitude also indicates the contribution from additional influencing factors.

- How does 2018 SSW differ from the other major SSW events in this regard?

  To address this question, we show $\Delta\mathrm{PRCP}_{\mathrm{Iberia}}$ (full value and the SSW-induced component) for individual historical SSW events from ERA5 (Fig. 1a in this reply). The 2018 SSW's $\Delta\mathrm{PRCP}_{\mathrm{Iberia}}$

ranks as the 5th strongest among all historical events (blue bars in Fig. 1a), but the SSW-induced component for the 2018 SSW does not stand out as one of the strongest compared to the other events (orange bars in Fig. 1a). This indicates a relatively small stratospheric signal in $\Delta PRCP_{Iberia}$ following the 2018 SSW. We further calculate the ratio of the magnitude of the SSW-induced component versus the magnitude of the residuals of $\Delta PRCP_{Iberia}$ for 46 historical SSW events from ERA5 (Fig. 1b). As can be seen, the 2018 SSW event falls within the bottom 25th percentile of all historical events, indicating that stratospheric variability plays a relatively small role in $\Delta PRCP_{Iberia}$ following the 2018 SSW compared to the other major SSW events.

In the revised manuscript, we have added a discussion on this to the conclusion section. Please see lines 532-537 of the revised manuscript, where we write:

*"This suggests that the exceptionally strong surface anomalies following the 2018 SSW are probably significantly driven by tropospheric variability. In fact, while the 2018 SSW ranks among the strongest on record in terms of the subsequent surface anomalies, the sudden warming itself is not particularly strong (Fig. S19). The contrast between the case's moderate intensity and its disproportionately strong surface impact also points to the substantial role of tropospheric variability in driving the observed surface anomalies following the 2018 SSW."*

**Comment 4:** The study highlights an apparent discrepancy in the timing of forecasted vs. observed precipitation anomalies over Iberia. Specifically, the forecasts tend to predict wet conditions immediately following the SSW, whereas in reality, the rainfall peak occurred approximately 15 days later. The authors argue that the surface impact of SSW occurs sooner in S2S models than in the reanalysis, suggesting that the timing of the transition to a negative NAO that occurred at the end of February 2018 was not predictably connected to the stratospheric anomalies, at least on subseasonal timescales. At the same time, the authors mention that persistent negative NAO-like SLP anomaly is a canonical surface response to SSW. Could the authors explore potential physical mechanisms behind this discrepancy that leads to different timing?

*Reply:* It has been pointed out by a previous study that a sequence of different weather regimes in the Euro-Atlantic area follows the 2018 SSW (Fig. 2 in this reply). These include positive NAO at the onset of the 2018 SSW, Scandinavian blocking in mid to late February, and negative NAO throughout most of March. Among the three regimes, the first two regimes (positive NAO and Scandinavian blocking) are atypical in the SSW-aftermath phase and therefore are unlikely driven by stratospheric variability associated with SSW. By contrast, the negative NAO is a canonical response to SSW, which is very likely driven by stratospheric variability. Since our analysis focuses on ensemble means that extract the SSW-induced variability, it is unsurprising that only the negative NAO is present in the ensemble means.

We therefore examine individual ensemble members to see if they envelope the observed evolution. To this end, for the late free ensemble of every SNAPSI model, we show $\Delta SLP_{Atlantic}$ in each of the 50 ensemble members (color lines in Fig. 3 in this reply). Here, the late free ensemble is shown as a supplement to Fig. 10 in the manuscript, of which the right panels show the multi-model ensemble mean of the late free ensemble. It turns out that, for most SNAPSI models, individual ensemble members envelope the observed evolution of the Atlantic SLP anomaly (black lines in Fig. 3 in this reply) throughout most of the period of interest. Hence, there is no evidence that these SNAPSI models are systematically biased in capturing the timing of shift to a negative NAO. An exception is GLOBO (Fig. 3b in this reply), for which the observed response falls outside of the envelope of individual ensemble members throughout most of the period of interest.

To sum up, the discrepancy in the timing of forecasted vs. observed anomalies arises because the ensemble means represent the SSW-induced variability, but not tropospheric internal variability. Looking at individual ensemble members, it turns out that most models' spread encompasses the observed response, suggesting that most SNAPSI models are not systematically biased in capturing the evolution of surface anomalies after the 2018 SSW.

Based on the above results, we have added a discussion to the revised manuscript. Please see lines 494-506 of the revised manuscript, where we write:

*"We then examine the evolution of the Atlantic SLP anomaly in individual ensemble members because the Atlantic SLP anomaly is a good predictor of the Iberian precipitation anomaly. To this end, for the late*

*free ensemble of every SNAPSI model, we show the daily Atlantic SLP anomaly over days [1,25] from each of the 50 ensemble members (color lines in Fig. 13). Here, the result of the late free ensemble is shown as a supplement to Fig. 10, of which the right panels show the multi-model ensemble mean of the late free ensemble. It turns out that, for most SNAPSI models, individual ensemble members envelope the observed evolution of the Atlantic SLP anomaly (black lines in Fig. 13) throughout most of the period of interest. Hence, there is no evidence that these SNAPSI models are systematically biased in capturing the timing of shift to a negative NAO. Again, an exception is GLOBO (Fig. 13b), for which the observed response falls outside of the envelope of individual ensemble members throughout most of the period of interest."*

*"These results suggest that for most models (except GLOBO), several individual ensemble members predict Iberian precipitation anomalies comparable to reanalysis. The observed evolution of the Atlantic SLP anomaly is also enveloped by individual ensemble members in those models, indicating that most SNAPSI models are not systematically biased in capturing the magnitude and timing of surface anomalies following the 2018 SSW."*

**Comment 5:** The manuscript discusses inter-model differences in the magnitude of forecasted precipitation anomalies. However, a more detailed exploration of the model-specific biases would be helpful. Are certain models systematically over- or under-predicting precipitation in the presence of stratospheric anomalies? Why CNRM-CM61 model exhibits mostly different behaviour than MMM?

*Reply:*

- Are certain models systematically over- or under-predicting precipitation in the presence of stratospheric anomalies?

  To address this question, we show $\Delta PRCP_{Iberia}$ in individual ensemble members for each set of forecast ensembles (blue whiskers in Fig. 4 in this reply). As can be seen, for seven of the eight SNAPSI models analyzed in this study, several individual ensemble members predict Iberian precipitation anomalies comparable to reanalysis (blue whiskers encompass ERA5 in Fig. 4). These results suggest that most SNAPSI models are not systematically biased in capturing the magnitude of the Iberian precipitation anomaly. The only exception is GLOBO, for which none of the ensemble members capture the observed magnitude of Iberian precipitation anomaly, regardless of the initialization dates or the application of the nudging technique (blue whiskers remain below ERA5). These results suggest that GLOBO is the only model that systematically under-predicts Iberian precipitation anomaly in the presence of stratospheric anomalies.

  Based on the above results, we have added a discussion to the revised manuscript. Please see lines 468-482 of the revised manuscript, where we write:

  *"Till now, the majority of the analysis focuses on ensemble means, in which some specific characteristics of the 2018 SSW are not present. For example, the ensemble mean $\Delta PRCP_{Iberia}$ is only about one-fourth as strong as the observed $\Delta PRCP_{Iberia}$ (Table 3). The timing of the ensemble mean vs. observed shift to strong, persistent negative NAO also exhibits an apparent discrepancy, with the former occurring almost right after the onset of the SSW, while the latter occurred approximately 15 days after the onset of SSW (Fig. 10). In this section, we will examine the ensemble spread of individual SNAPSI models to see if some members can capture the magnitude and timing of the observed tropospheric anomalies after the 2018 SSW. This also allows us to find out whether certain SNAPSI models are systematically biased in representing these characteristics of the 2018 SSW."*

  *"We start with the Iberian precipitation anomaly because it is the focus of our study. For each set of forecast ensembles, we show the most extreme data value (among the 50 ensemble members) (blue whiskers in Fig. 11). As can be seen, for seven of the eight SNAPSI models analyzed in this study, some ensemble members capture the observed magnitude of the Iberian precipitation anomaly (blue whiskers encompass ERA5 in Fig. 11). The only exception is GLOBO, for which none of the ensemble members capture the observed magnitude of the Iberian precipitation anomaly, regardless of the initialization dates or the application of the nudging. These results indicate that GLOBO is the only model that systematically underestimates the magnitude of the Iberian precipitation anomaly."*

- Why CNRM-CM61 model exhibits mostly different behaviour than MMM?

We're unsure which particular aspect the reviewer is referring to because CNRM-CM61 exhibits different behaviour than MMM in a couple of fields. So here, a couple of points are listed:

(a) Regarding the early free ensemble, CNRM-CM61 is the only model that forecasted a neutral stratosphere state, whereas all the other models forecasted a strong stratospheric cooling (Fig. 7a in the manuscript).

Ongoing investigations by another SNAPSI working group are expected to address this topic in more detail, with a manuscript to be submitted shortly. We refer the reviewer to future papers in this Special issue: Stratospheric impacts on climate variability and predictability in nudging experiments (WCD/GMD inter-journal SI).

(b) Regarding the late free ensemble, CNRM-CM61's $\Delta \text{SLP}_{\text{Arctic}}$ and $\Delta \text{PRCP}_{\text{Scandinavia}}$ have much larger magnitude than the other models (Fig. 8c in the manuscript).

This might arise, at least partially, from the fact that CNRM-CM61's late free ensemble forecasted the strongest stratospheric warming amongst the eight SNAPSI models (Fig. 8a in the manuscript).

**Comment 6:** Given the underestimation of precipitation anomalies in the model ensembles, is there a potential for bias correction techniques or ensemble calibration methods to enhance forecast skill? How the findings of the paper can be used in hydrological prediction?

*Reply:*

- is there a potential for bias correction techniques or ensemble calibration methods to enhance forecast skill?

  To address this question, we show $\Delta \text{PRCP}_{\text{Iberia}}$ in individual ensemble members for each set of forecast ensembles (blue whiskers in Fig. 4 in this reply). As can be seen, for seven of the eight SNAPSI models analyzed in this study, several individual ensemble members predict Iberian precipitation anomalies comparable to reanalysis (blue whiskers encompass ERA5 in Fig. 4). These results suggest that most SNAPSI models are not systematically biased and thus may not require a bias correction (see our reply to Comment 5). In addition, we are unable to fully assess precipitation biases because hindcast climatologies for the SNAPSI model versions are not available.

- How can the findings of the paper be used in hydrological prediction?

  Here are a couple of examples of how our findings can be leveraged in hydrological prediction:

  (a) We find that SSW can significantly increase the risk of Iberian precipitation extremes (nudged vs. control ensembles). Therefore, by identifying an SSW event, hydrological forecasters could adjust the ensemble river flow predictions toward a higher risk of floods in certain basins in southern Europe.

  (b) We find that the subseasonal predictability of precipitation signals after the 2018 SSW arises from a successful forecast of the sudden warming itself. Therefore, when designing ensemble forecasts, initializing forecast ensembles with accurate stratospheric conditions (e.g., during SSW events) may provide more skillful and confident hydrological predictions.

  (c) We find that the stratosphere represents an important source of S2S predictability for precipitation over Europe. Therefore, stratospheric diagnostics (e.g., SPV strength) can be used as predictors in statistical or machine learning models to improve the skill of flood/drought outlooks.

**Minor comments**

**Comment 1:** L14-15 The idea of this sentence is not clear to the reader: 'Nonetheless, the likelihood of Iberian rainfall extremes comparable to or even stronger than the one observed doubles in the nudged ensemble, compared to the control ensemble.' Please rewrite this sentence, possibly dividing it into two.

*Reply:* In the revised manuscript, this sentence has been rewritten. Please see lines 14-15 of the revised manuscript, where we write:

*"Nonetheless, Iberian rainfall extremes of equal strength or stronger than the one observed are twice as likely in the nudged ensemble than in the control ensemble."*

**Comment 2:** L40 The authors state: 'There is thus a clear need to evaluate the capabilities of state-of-the-art operational S2S models to predict such an SSW'. However, while the statement is true, there are several works on this topic that have already addressed this issue which should be cited here.

***Reply:*** In the revised manuscript, we have cited several works on this topic. Please see lines 40-43 of the revised manuscript, where we write:

*"There is thus a clear need to evaluate the capabilities of state-of-the-art operational S2S models to predict such an SSW and the extreme tropospheric state after it. The 2018 SSW event provides an excellent case study with which to conduct the evaluation (Karpechko et al., 2018; Rao et al., 2020; Butler et al., 2020)."*

**Comment 3:** Table 1 The manuscript currently refers to model versions in the main text but uses the corresponding participating center names in the figures. This inconsistency may cause confusion for readers. It is customary to use the participating center name as the model name. Therefore, for clarity and consistency, it is better to refer to models by their participating center names, as done in the figures.

***Reply:*** There is consensus among all SNAPSI working groups to refer to model versions in the main text. In this study, model versions are also used in figures in the main text (see Figs. 7-8 in the main text).

**Comment 4:** Table 2 It would be beneficial to add information on nudging levels (90 hPa) for both nudged and control experiments.

***Reply:*** In the revised manuscript, we have added information on nudging levels to Table 2. Please see the caption of Table 2 in the revised manuscript, where we write:

*"For both nudged and control ensembles, the nudging region has a lower limit of 90 hPa."*

**Comment 5:** L113-115 The authors state that they remove the seasonal cycle to calculate ERA5 anomalies. But in L120 they state that 'the anomalies in ERA5 are also calculated relative to the climatological state' as for the IMERG and GSN data. Does climatological state mean the same as the seasonal cycle here? This is confusing, could you please make the description of anomalies calculation more precise?

***Reply:*** I apologize for the confusion. In this study, the term 'climatological state' does not mean the same as the term 'seasonal cycle'.

- The climatological state refers to the daily climatology computed as the multi-year average for each calendar day.

- The seasonal cycle is a smoothed version of the daily climatology. In this study, the seasonal cycle is defined as the mean and first three Fourier harmonics of the daily climatology.

As the reviewer has noticed, the unsmoothed daily climatology is used to calculate anomalies for both IMERG and GSN data, as well as ERA5 precipitation that is directly compared to IMERG and GSN. This is because the observational data like IMERG and GSN have missing values on certain days and at certain locations. These missing values may result in gaps in the daily climatology at certain locations. The presence of gaps in the daily climatology precludes the use of smoothing techniques like the Fourier filter.

In the revised manuscript, we have reorganized part of section 2.3 and section 2.4. In particular, both the seasonal cycle and daily climatology have been defined, and their differences have been highlighted. Please see section 2.3 and section 2.4 of the revised manuscript.

**Comment 6:** L142-143 Comparison with Ayarzaguena et al., 2018 is a repetition from Introduction L50. I think this should be avoided, however, this also highlights the importance to clearly explain the additional value of observations (see Major comments 1).

***Reply:*** In the revised manuscript, we have removed the comparison and explained the value of observations. Please see lines 146-149 of the revised manuscript, where we write:

*"In this section, we use NASA satellite observations and GSN in situ station observations of precipitation to identify precipitation signals after the 2018 SSW, and then compare them to those in ERA5 precipitation. The inclusion of observational precipitation adds value by providing an independent and often more accurate reference, helping to validate and complement ERA5 precipitation that does not directly assimilate any rain-gauge data (Lavers et al., 2022)."*

**Comment 7:** Figure 1a It would be good to add a short explanation of why there is areas with no data to the caption.

**Reply:** In the revised manuscript, we have added an explanation to the caption of Fig. 1. Please see the caption of Fig. 1 in the revised manuscript, at the end of the caption we write:

*"Note that IMERG precipitation in panel a provides only partial spatial coverage at latitudes above $60°$. This is because infrared-based precipitation estimates cannot be included at higher latitudes, so the coverage is limited to grid boxes for which there is no snow/ice on the surface."*

**Comment 8:** Figures 2 and 4 The manuscript briefly discusses precipitation anomalies over North America (L158-159, L212-214), but it also states that these signals are not directly attributable to the 2018 SSW and are not the focus of the analysis. Given this, their inclusion in the main text may distract from the core findings. To improve clarity and maintain focus, I suggest moving the discussion of North American precipitation signals to the Supplementary Material.

**Reply:** In the revised manuscript, the discussion of North American precipitation signals has been moved to the Supplementary Material. Please see Figures 2 and 4 in the revised manuscript, as well as Supplementary Figures 2 and 7.

**Comment 9:** L266-270 Not all SSWs lead to negative NAO and precipitation anomalies. I suggest that the authors add a discussion on how this variability might affect the results, particularly in terms of predictability and the generalizability of the conclusions. Addressing this point would strengthen the interpretation of the findings.

**Reply:** A discussion has been added to the revised manuscript. Please see lines 275-279 of the revised manuscript, where we write:

*"It is important to note that not all SSW events lead to a persistent negative NAO or European precipitation anomalies. This event-to-event variability in the surface response to SSWs introduces uncertainty into any predictive framework based solely on the occurrence of SSWs. One way to mitigate this uncertainty is by conditioning predictions on specific characteristics of the SSW (e.g., type, magnitude, and tropospheric precursors) (Maycock and Hitchcock, 2015; Kodera et al., 2016; Runde et al., 2016; de la Cãmara et al., 2017; White et al., 2019; Xu et al., 2022)."*

**References**

Ayarzaguena, B., Barriopedro, D., Perez, J. M. G., Abalos, M., de la Camara, A., Herrera, R. G., Calvo, N., and Ordonez, C.: Stratospheric connection to the abrupt end of the 2016/2017 iberian drought, Geophysical Research Letters, 45, 12,639–12,646, 2018.

Butler, A. H., Lawrence, Z. D., Lee, S. H., Lillo, S. P., and Long, C. S.: Differences between the 2018 and 2019 stratospheric polar vortex split events, Quarterly Journal of the Royal Meteorological Society, 146, 3503–3521, https://doi.org/https://doi.org/10.1002/qj.3858, 2020.

de la Cãmara, A., Albers, J., Birner, T., Garcia, R. R., Hitchcock, P., Kinnison, D. E., and Smith, A. K.: Sensitivity of sudden stratospheric warmings to previous stratospheric conditions, Journal of the Atmospheric Sciences, 74, 2857–2877, 2017.

Karpechko, A. Y., Perez, A. C., Balmaseda, M., Tyrrell, N., and Vitart, F.: Predicting sudden stratospheric

warming 2018 and its climate impacts with a multi-model ensemble, Geophysical Research Letters, p. 2018GL081091, 2018.

Kodera, K., Mukougawa, H., Maury, P., Ueda, M., and Claud, C.: Absorbing and reflecting sudden stratospheric warming events and their relationship with tropospheric circulation, Journal of Geophysical Research: Atmospheres, 121, 80–94, 2016.

Lavers, D. A., Simmons, A., Vamborg, F., and Rodwell, M. J.: An evaluation of ERA5 precipitation for climate monitoring, Quarterly Journal of the Royal Meteorological Society, 148, 3152–3165, https://doi.org/https://doi.org/10.1002/qj.4351, 2022.

Maycock, A. C. and Hitchcock, P.: Do split and displacement sudden stratospheric warmings have different annular mode signatures?, Geophysical Research Letters, 42, 10 943–10 951, 2015.

Rao, J., Garfinkel, C. I., and White, I. P.: Predicting the Downward and Surface Influence of the February 2018 and January 2019 Sudden Stratospheric Warming Events in Subseasonal to Seasonal (S2S) Models, Journal of Geophysical Research: Atmospheres, 125, e2019JD031 919, https://doi.org/https://doi.org/10.1029/2019JD031919, e2019JD031919 2019JD031919, 2020.

Runde, T., Dameris, M., Garny, H., and Kinnison, D. E.: Classification of stratospheric extreme events according to their downward propagation to the troposphere, Geophysical Research Letters, 43, 6665–6672, 2016.

White, I., Garfinkel, C. I., Gerber, E. P., Jucker, M., Aquila, V., and Oman, L. D.: The downward influence of sudden stratospheric warmings: Association with tropospheric precursors, Journal of Climate, 32, 85–108, 2019.

Xu, Q., Chen, W., and Song, L.: Two Leading Modes in the Evolution of Major Sudden Stratospheric Warmings and Their Distinctive Surface Influence, Geophysical Research Letters, 49, e2021GL095 431, https://doi.org/https://doi.org/10.1029/2021GL095431, e2021GL095431 2021GL095431, 2022.

[Figure]

Figure 1: (a) $\Delta PRCP_{Iberia}$ for individual historical SSW events from ERA5 (bars). For each SSW event, the blue and orange bars indicate the full value and the SSW-induced component of $\Delta PRCP_{Iberia}$, respectively. The black horizontal line indicates the full value of $\Delta PRCP_{Iberia}$ for the 2018 SSW event. (b) The ratio of the magnitude of the SSW-induced component versus the magnitude of the residuals of $\Delta PRCP_{Iberia}$ for historical SSW events from ERA5. On the box, the red line is the median, the edges of the box are the 25th and 75th percentiles, and the whiskers extend to the most extreme data value that is not an outlier. The blue circle indicates the value of the ratio for the 2018 SSW event. Here, the SSW-induced component and the residuals are obtained using the historical regression from ERA5 (see equations 4-5 in the manuscript).

[Figure]

Figure 2: Daily evolution of weather regimes (color bars). The daily sequence of WRs over the Euro-Atlantic area from 1 January to 31 March 2018. Figure 1c from Ayarzaguena et al. (2018).

[Figure]

Figure 3: Time series of $\Delta\mathrm{SLP}_{\mathrm{Atlantic}}$ over days [1,25] after the 2018 SSW from ERA5 (thick black curve) and from the late free ensemble (thin color curves and thick white curve). For the late free ensemble, the thin color curves indicate 50 ensemble members for each model, and then the thick white curve indicates the ensemble mean of 50 members. Grey and blue shading correspond to 1 and 2 standard deviations around the multi-member average. Here, a 5-day moving average is applied to $\Delta\mathrm{SLP}_{\mathrm{Atlantic}}$ to reduce high-frequency variations.

[Figure]

Figure 4: $\Delta\mathrm{PRCP}_{\mathrm{Iberia}}$ by individual SNAPSI models. For each ensemble, the color marker is the multi-member average, the black whiskers correspond to 1 standard deviation around the multi-member average, and the blue whiskers extend to the most extreme data value. F0125: early free ensemble; F0208: late free ensemble; N0125: early nudged ensemble; N0208: late nudged ensemble. The solid black horizontal line indicates $\Delta\mathrm{PRCP}_{\mathrm{Iberia}}$ from ERA5.